# Enhancing One-Shot Pruned Generative Pre-training Language Models through Sparse-Dense-Sparse Mechanism

## Abstract

Generative pre-trained language models (PLMs) are engineered to be robust in contextual understanding and exhibit outstanding performance in various natural language processing tasks. However, their considerable size incurs significant computational and storage costs. Modern pruning strategies employ one-shot techniques to compress PLMs without the need for retraining on task-specific or otherwise general data; however, these approaches often lead to an indispensable reduction in performance. In this paper, we propose **SDS**, a **S**parse-**D**ense-**S**parse pruning framework to enhance the performance of the pruned PLMs from a weight distribution optimization perspective. We outline the pruning process in three steps. Initially, we prune less critical connections in the model with conventional one-shot pruning methods. Next, we reconstruct a dense model featuring a pruning-friendly weight distribution by reactivating pruned connections with *sparse regularization*. Finally, we perform a second pruning round, yielding a superior pruned model compared to the initial pruning. Notably, SDS requires only a limited number of calibration samples, comparable to typical one-shot pruning methods, but significantly outperforms them. Experimental results demonstrate that, under an identical sparsity configuration, SDS outperforms the state-of-the-art pruning technique SparseGPT and Wanda by decreasing language comprehension perplexity by an average of 6.4 and achieving an average accuracy improvement of 1.9% across multiple downstream tasks on OPT, GPT, and LLaMA.

## 1 Introduction

Generative pre-trained language models (PLMs) (Vaswani et al., 2017), such as ChatGPT, have revolutionized various applications in natural language processing. However, the considerable size of PLMs results in notable drawbacks such as increased latency and energy consumption. Compression methods for vision models such as convolutional neural networks, which perform *pre-training, compression, and fine-tuning* workflow with quantization or pruning (Liang et al., 2021), are ill-suited for PLMs due to their prohibitive training cost.

Recent pruning research, such as SparseGPT (Frantar & Alistarh, 2023) and Wanda (Sun et al., 2023), has introduced effective one-shot compression techniques for PLMs. These methods can compress up to 50% of the parameters in the fully-connected layers of large PLMs with negligible impact on performance. However, the effectiveness diminishes when applied to undersized ones, which are usually more fully trained and are considered difficult to compress. For instance, SparseGPT, the state-of-the-art pruning method, yields a perplexity of 31.58 when applied to prune 50% of the weights in OPT-350m. This score is worse than the 27.66 perplexity observed in OPT-125m, a smaller dense model with roughly half the parameters of OPT-350m. Furthermore, when stricter sparsity constraints are employed, such as 2:4 or 4:8 sparse configurations (Mishra et al., 2021) for computational acceleration, the performance deteriorates even further. Therefore, it is essential to optimize the poorly pruned PLMs.

From a macroscopic point of view, PLMs are not designed to be aware of subsequent pruning since they lack pruning-related regularization during pre-training. As a result, pruning PLMs while maintaining their performance proves challenging, especially for undersized ones.

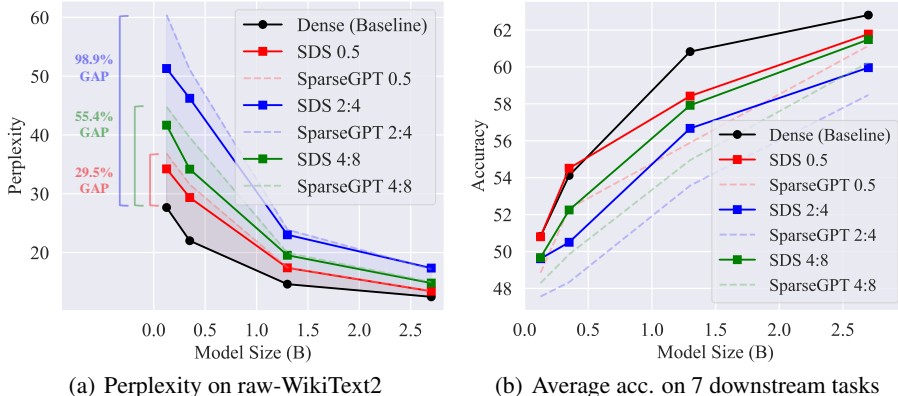

(a) Perplexity on raw-WikiText2  (b) Average acc. on 7 downstream tasks

Figure 1: SDS compensates for the performance drop in SparseGPT-pruned, undersized OPTs.

Neurons in the human brain show sparse-to-dense-to-sparse connectivity as they grow (Herculano-Houzel et al., 2010). This observation inspired us to perform a sparse-to-dense-to-sparse process to achieve a better pruning scheme that benefits from sparse regularization. We preliminarily explored layer-by-layer dense reconstruction to find a performance upper bound. Intriguingly, we discovered that sparse models could bounce back to performance levels equivalent to their dense counterparts using only a few unlabeled samples (cf., Table 1). It reveals two key insights: first, pruned PLMs can be optimized with limited resources; second, the amount of knowledge lost during the pruning process is negligible. These insights lay the foundation for the work presented in this paper.

We propose a three-step **S**parse-**D**ense-**S**parse (**SDS**) pruning framework to enhance the performance of pruned generative pre-trained language models. **In the first step**, we employ conventional one-shot pruning methods on a PLM to remove irrelevant connections. **In the second step**, we perform a dense reconstruction of the sparse model to reactivate the pruned connections, aiming to identify a dense model with enhanced pruning awareness. This process is aided by a multidimensional **sparse regularization** strategy, which optimally guides the weight distribution, rendering it more pruning-friendly for the subsequent step. **In the third step**, we apply SparseGPT to further prune and adjust the weights of the re-dense model. Importantly, SDS requires only a limited number of unlabeled samples for calibration, identical to conventional one-shot methods. Experimental results demonstrate that SDS outperforms SparseGPT and Wanda under the same sparsity configuration, reducing the average perplexity by 6.4 and improving accuracy by 1.9% across multiple downstream tasks in OPT, GPT, and LLaMA (cf., Figure 1). The main contributions of the paper are summarized as follows:

- We introduce SDS, a three-step sparse-dense-sparse framework. It involves weight redistribution and pruning, enhancing the performance of the one-shot pruned generative pre-trained language models.
- We design sparse regularization strategies that improve the effectiveness of re-dense weight reconstruction as well as find a more pruning-friendly weight distribution.
- Experimental results demonstrate that SDS sets a new state-of-the-art in both language comprehension and downstream task performance.

Table 1: **Pruning PLMs incurs minimal knowledge loss.** We apply 2:4 sparse to PLMs with SparseGPT, and their performance decreases in the raw-WikiText2 task. However, upon reactivating the sparse weights layer by layer with only 128 samples from C4, a substantial performance improvement is observed.

| PLMs | Dense | 2:4 Sparse | Re-dense |
|---|---|---|---|
| OPT-125m | 27.66 | 60.43 | 27.94 |
| OPT-350m | 22.01 | 51.11 | 22.25 |

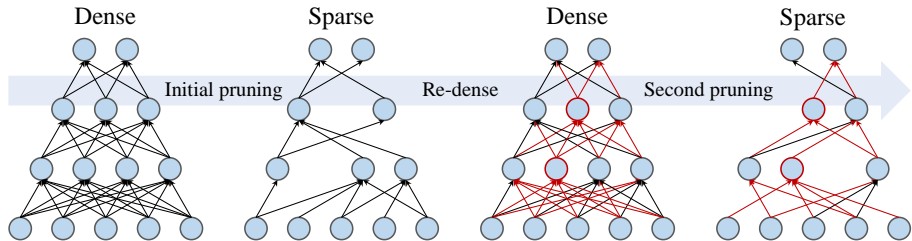

Figure 2: An overview of the steps of the SDS framework, which is divided into initial pruning, re-dense weight reconstruction, and a second round of pruning.

## 2 SDS: ENHANCING ONE-SHOT PRUNING THROUGH SPARSE-DENSE-SPARSE MECHANISM

This section presents the Sparse-Dense-Sparse (SDS) framework to perform one-shot pruning for generative pre-trained language models (PLMs). First, we provide a brief overview of the core Transformer architecture, which is fundamental to most PLMs. A standard Transformer block consists of two main modules: a multi-head attention (MHA) layer and a feed-forward network (FFN). Let $X_{l-1} \in \mathbb{R}^{n \times d}$ represent the input of the $l$-th Transformer block, where $n$ is the sequence length, and $d$ is the size of the hidden state. The block output $X_l \in \mathbb{R}^{n \times d}$ can be formulated as:

$$X = \mathrm{MHA}\left(\mathrm{LayerNorm}\left(X_{l-1}\right)\right) + X_{l-1}, \quad X_l = \mathrm{FFN}(\mathrm{LayerNorm}\left(X\right)) + X. \quad (1)$$

MHA consists of $h$ heads, represented as $W^O \cdot \mathrm{Concat}(\mathrm{head}_1, \mathrm{head}_2, \ldots, \mathrm{head}_h)$, with $W^O$ responsible for the output projection. Furthermore, the $i$-th head can be expressed as:

$$\mathrm{head}_i = \mathrm{Attention}([W^Q X]_i, [W^K X]_i, [W^V X]_i, M), \quad (2)$$

$$\mathrm{Attention}(Q, K, V, M) = \mathrm{softmax}\left(M \odot \frac{QK^\top}{\sqrt{d_k}}\right) V, \quad (3)$$

where $Q$, $K$, and $V$ represent the query, key, and value sequences, respectively, and their corresponding projection weights are $W^Q$, $W^K$, and $W^V$. $d_k$ is the dimension of the key vectors, and $M$ is the mask matrix to selectively ignore or give weight to certain tokens in the input sequence. FFN expands and contracts input dimensions through hidden layers, introducing non-linearities to enhance representation learning, which consists of two fully-connected layers, with their weights represented as $W^{FC1}$, $W^{FC2}$, respectively.

In this paper, we focus on pruning the weights in fully-connected layers, which are emphasized by $W^Q$, $W^K$, $W^V$, $W^O$, $W^{FC1}$ and $W^{FC2}$ from the outset.

The SDS framework consists of three steps: initial pruning (Section 2.1), re-dense weight reconstruction (Section 2.2), and a second round of pruning (Section 2.3). By optimizing weight distribution through these steps, the SDS framework enhances the performance of pruned PLMs. The overall process of the SDS framework and the evolution of weight distribution during its steps are depicted in Figures 2 and 3, respectively.

### 2.1 INITIAL PRUNING

The SDS framework initiates by eliminating the less important connections in PLMs using conventional one-shot pruning methods. SparseGPT (Frantar & Alistarh, 2023) leverages second-order information for guiding sparse mask selection and modifying weights. Concretely, given a particular sparsity, SparseGPT compensates for the error that occurs during the pruning of the $c$-th column of the weight matrix $W^{\mathrm{dense}}$ by modifying the weights of subsequent columns (here we use subscripts for row and column indexes, omitting the layer number):

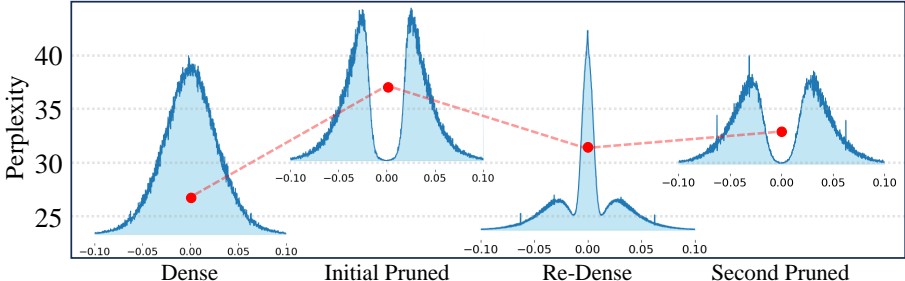

Figure 3: **The evolution of weight distribution within the SDS framework**. The weights are extracted from the FFN in the 12-th transformer block of OPT-125m, with a sparsity level of 50%. Initially, the dense weights follow a Gaussian distribution. After being pruned by SparseGPT, a concentrated, bimodal distribution emerges (zero values are omitted in sparse weight distributions for better clarity). Followed by connection reconstruction with sparse regularization, a three-peaked distribution materializes. Finally, the second pruning round attenuates the sharp peaks, resulting in a softer bimodal distribution. The red dotted line represents the perplexity on raw-WikiText2. The second pruned model achieves a lower perplexity than the initially pruned one.

$$W_{:,c}^{\text{sparse}} = \mathbb{1}\left(\text{sort}\left(\frac{W_{:,c}^{\text{dense}\,2}}{\left[H^{-1}\right]_{c,c}^{2}}\right) > \text{sparsity}\right) \odot W_{:,c}^{\text{dense}}, \tag{4}$$

$$W_{:,c+1:}^{\text{dense}} = W_{:,c+1:}^{\text{dense}} - \frac{\left(W_{:,c}^{\text{dense}} - W_{:,c}^{\text{sparse}}\right)^{2}}{\left[H^{-1}\right]_{c,c}^{2}} \cdot H_{:,c}^{-1}, \tag{5}$$

where $H = XX^{\top}$ is the Hessian matrix, which identifies directions on the model's loss surface that have minimal curvature, corresponding to weights that minimally impact the performance.

SparseGPT demonstrates robust performance on over-parameterized models like OPT-175B, achieving negligible performance degradation. However, its efficacy diminishes when applied to under-sized (more fully trained) ones. This limitation may arise from the loss of the optimization information due to the second-order-only and the Hessian matrix diagonal-only assumption. Concurrently, the weights in the latter columns face more accumulated errors than the previous ones, leading to an unbalanced optimization. Generally, during the pre-training phase, there is no sparse regularization applied to PLMs. This means that the model lacks awareness of the subsequent pruning process.

We incorporate SparseGPT into the SDS framework as the initial step (see Section A.5 for SDS with the Wanda-based pruning method). In the subsequent steps, we aim to identify a superior sparse model from the perspective of weight distribution optimization.

## 2.2 RE-DENSE WEIGHT RECONSTRUCTION

Table 1 demonstrates that there is more than one possible set of dense-weight solutions with similar performance. To this end, our goal is to find a weight solution that exhibits pruning awareness and forms **a pruning-friendly dense model** to serve as a new starting point for pruning. Concretely, we implement layer-by-layer knowledge distillation with limited unlabeled samples to reactivate the connections in pruned PLMs. This method guarantees high efficiency while preserving the multitasking capabilities of the re-dense PLMs.

However, a naïve re-dense weight reconstruction is insufficient. To circumvent ending up with a re-dense solution resembling the original dense model, we introduce three sparse regularization strategies. **a) Sparse inherited traits**: the initial pruning cannot be omitted; it provides prior information about which weights are important for the re-dense weight reconstruction process. **b) Data-based regularization**: hard-to-learn samples are used as the inputs of re-dense weight reconstruction to avoid overfitting aberration. **c) Weight-based regularization**: typical weight regularization is also employed to endow the re-dense weights with sparse features, thereby directly increasing the prun-

ing friendliness. We choose L1 regularization (Tibshirani, 1996) and L2 regularization, commonly referred to as weight decay (Loshchilov & Hutter, 2019), to meet the requirement.

According to the above three considerations, the re-dense weight reconstruction process is specified in the following. Given the original dense weight $\boldsymbol{W}_l^{\text{dense}}$ in layer $l$, the sparse weight $\boldsymbol{W}_l^{\text{sparse}}$ from the initial pruning step, and $\boldsymbol{X}_{l-1}$ collected during the forward propagation of the sparse model, the re-dense weight $\widehat{\boldsymbol{W}}_l^{\text{re-dense}}$ is obtained by:

$$\widehat{\boldsymbol{W}}_l^{\text{re-dense}} = \text{argmin}_{\boldsymbol{W}_l^{\text{sparse}}} \left( \left\| \boldsymbol{W}_l^{\text{dense}} \boldsymbol{X}_{l-1} - \boldsymbol{W}_l^{\text{sparse}} \boldsymbol{X}_{l-1} \right\|_2^2 + \lambda_1 \|\boldsymbol{W}_l^{\text{sparse}}\|_1 + \lambda_2 \|\boldsymbol{W}_l^{\text{sparse}}\|_2^2 \right), \tag{6}$$

where $\lambda_1$ and $\lambda_2$ are used to control the ratio of the L1 and L2 regularization, they are set to be 0.1 by default. The distribution of $\widehat{\boldsymbol{W}}_l^{\text{re-dense}}$ are shown in Figure 3. Firstly, the parameters obtained through re-dense weight reconstruction show a clear three-peaked distribution. This distribution displays a higher sharpness around zero than the original dense model, which is a terrific phenomenon. It indicates that weights with lower norms suppress irrelevant information for learning, a trait referred to as pruning-friendliness (Han et al., 2017). Secondly, the re-dense weight reconstruction yields a model with performance slightly below that of the original dense model but significantly better than the initial sparse model, aligning with our expectations. On the one hand, we only use a small amount of unlabeled data during the reconstruction phase. On the other hand, the input $\boldsymbol{X}_{l-1}$ is generated by the sparse model, which is challenging to optimize, thereby avoiding overfitting on simple samples. There is a more detailed analysis in A.2.

## 2.3 SECOND PRUNING: SPARSE WEIGHT ADJUSTMENT

Directly adjusting weights in a sparse-to-sparse manner seems intuitive for enhancing a sparse model's performance; however, when applied to a model after the initial pruning stage, it only results in minor performance gains on the lightest models. To begin with, the weights that SparseGPT has pruned will show a sharp bimodal distribution. Such a stable and spiky distribution with low variance constrains the optimization potential. In addition, the benefits of knowledge distillation with unlabeled samples are not superior to those of task-specific fine-tuning.

Considering the aforementioned challenges, we introduce sparse weight adjusting as the concluding step in the SDS framework. Firstly, the re-dense model obtained with sparse regularization guidance will inevitably perform inferior to the pre-trained model. As a result, directly pruning it would not be ideal. Therefore, performance optimization of the second-pruned model is necessary. Secondly, after the re-dense model is pruned, its distribution tends to moderate, and thus there is more room for optimization. To elaborate, we first prune the re-dense model using the same method employed during the initial pruning, yielding $\boldsymbol{W}^{\text{sparse-2nd}}$. Subsequently, weight adjusting is conducted utilizing a soft sparse mask:

$$\widehat{\boldsymbol{W}}_l^{\text{SDS}} = \boldsymbol{M}_l^{\text{soft}} \odot \left( \text{argmin}_{\boldsymbol{W}_l^{\text{sparse-2nd}}} \left\| \boldsymbol{W}_l^{\text{re-dense}} \boldsymbol{X}_{l-1} - \boldsymbol{W}_l^{\text{sparse-2nd}} \boldsymbol{X}_{l-1} \right\|_2^2 \right), \tag{7}$$

where $\boldsymbol{X}_{l-1}$ is also collected from the forward propagation of the second pruned model. $\boldsymbol{M}_l^{\text{soft}}$ represents a soft sparse mask, which is dynamically selected by $|\boldsymbol{W}_l^{\text{sparse-2nd}}|$ in each iteration. Due to the inherent awareness of activation information from backpropagation, the magnitude (*absmin*) Hagiwara (1994) mask selection metric can achieve results similar to the Hessian metric in SparseGPT and the activation-aware metric in Wanda. In both steps of weight adjustment, the L2 loss is utilized, inherently emphasizing the loss in regions with outliers (Xiao et al., 2023), which plays a pivotal role in the performance of language models. Therefore, outliers can be protected and less affected by weight adjustments.

As shown in Figure 3, the weight presented after the second pruning will become moderate, i.e., the weight distribution of the second-pruned model is smoother and more uniform than that of the initial pruning step, which means that the model parameters have suitable values in different ranges, possessing a stronger ability to adapt to unseen data.

The overall process of the SDS framework is shown in Algorithm 1 in A.1.

## 3 EXPERIMENTS

### 3.1 EXPERIMENTAL SETTINGS

**Models.** We primarily work with the Open Pre-trained Transformers (OPTs) (Zhang et al., 2022). Among them, the 125M, 350M, 1.3B, and 2.7B versions are the undersized ones that we chiefly focus on. The modules to be pruned are the computationally intensive `self-attn.q_proj`, `self-attn.k_proj`, `self-attn.v_proj`, `self-attn.out_proj`, `fc1`, and `fc2` layers constructed from fully-connected layers (see Section A.5 for experiments on GPT and LLaMA).

**Calibration.** For the data used in calibration, we adhere to the approach outlined in SparseGPT, selecting 128 segments of 2048 tokens each from the initial partition of the C4 dataset (Raffel et al., 2020). This dataset, sourced from a broad array of internet text, guarantees that our experiments are zero-shot, as no task-specific information is exposed during our optimization process.

**Datasets and evaluation.** Regarding evaluation metrics, our primary emphasis is on perplexity, which remains a challenging and reliable metric well suited for evaluating the language modeling capability of compressed models (Frantar & Alistarh, 2022; Frantar et al., 2022; Yao et al., 2022). We consider raw-WikiText2 (Merity et al., 2017) test sets for perplexity validation. To explore the impact of compression on other downstream tasks, we also provide zero-shot accuracy results for COPA (Wang et al., 2019), Lambada (Paperno et al., 2016), OpenBookQA (Mihaylov et al., 2018), PIQA (Bisk et al., 2020), RTE (Wang et al., 2018), StoryCloze (Sharma et al., 2018) and Winogrande (Sakaguchi et al., 2019).

**Setup.** We implement the SDS framework in PyTorch (Paszke et al., 2019) and use the HuggingFace Transformers library (Wolf et al., 2020) for managing models and datasets. We follow SparseGPT to prune the pre-trained model in the initial pruning step. In the re-dense weight reconstruction step, we use 128 samples as inputs to perform layer-by-layer knowledge distillation: the number of distillation epochs is 200, the learning rate is 0.1, the loss function is L2 loss, and the regularization strategy contains L1 regularization and weight decay with a ratio of 0.1. After the current layer is reconstructed, the initial pruned layer's output is directly used as the input of the next layer, without an additional forward propagation to obtain the reconstructed layer output as the input of the next layer. In the second pruning step, we again use SparseGPT to prune the re-dense model and use the same configuration as in the previous step but without regularization to further adjust the weights of the pruned model with a soft sparse mask. The SDS framework uses the same samples throughout, meaning that there is no sample overload.

### 3.2 RESULTS

**Varying sparsity levels.** We conduct experiments with varying sparsity levels for unstructured pruning; the results are depicted in Figure 4. It can be observed that the Sparse-Dense-Sparse (SDS) framework is effective in optimizing the performance of pruned PLMs at either high or low sparsity.

**Performance on language modeling.** We focus on three sparsity configurations: 50% sparsity for model compression, 2:4 and 4:8 sparsity for both compression and real-world computational acceleration on specialized hardware. We conduct a comprehensive performance evaluation using raw-WikiText2, measuring perplexity for each step of the SDS framework, as shown in Table 2.

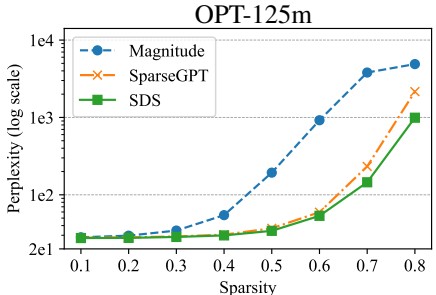 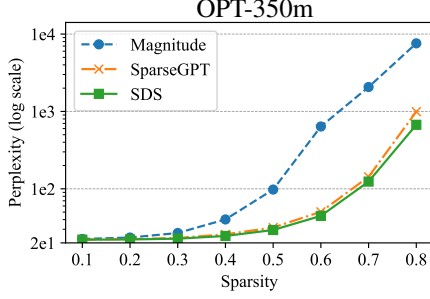

Figure 4: Sparsity vs. Perplexity in OPTs.

Table 2: **Perplexity on raw-WikiText2. S**DS represents the initially pruned model, which is also our baseline (SparseGPT). S**D**S represents the dense model obtained in the re-dense weight reconstruction step. SD**S** represents the model obtained in the second round of pruning.

| PLM | Dense | Sparsity | The workflow of the SDS framework | | |
|---|---|---|---|---|---|
| | | | **S**DS | S**D**S | SD**S** |
| OPT-125m | 27.66 | 50% | 36.85 | 31.78 | **34.23** |
| | | 2:4 | 60.43 | 44.46 | **51.30** |
| | | 4:8 | 44.77 | 37.82 | **41.66** |
| OPT-350m | 22.01 | 50% | 31.58 | 24.78 | **29.36** |
| | | 2:4 | 51.11 | 31.58 | **46.23** |
| | | 4:8 | 39.59 | 26.15 | **34.18** |
| OPT-1.3b | 14.62 | 50% | 17.46 | 17.39 | **17.40** |
| | | 2:4 | 23.90 | 20.00 | **23.02** |
| | | 4:8 | 19.95 | 18.06 | **19.54** |
| OPT-2.7b | 12.46 | 50% | 13.48 | 12.50 | **13.42** |
| | | 2:4 | **17.18** | 13.77 | 17.36 |
| | | 4:8 | 14.98 | 12.88 | **14.81** |

The superiority of the SDS framework is especially pronounced in OPT-125m and OPT-350m. Averaging across these models, SDS surpasses SparseGPT by 7.1% at 50% sparsity, 18.5% at 2:4, and 7.5% at 4:8 configurations. These results robustly affirm that SDS yields significantly superior performance over undersized models in various sparsity configurations. In OPT-1.3b and OPT-2.7b, the SDS framework still holds a performance over the SparseGPT baseline. Specifically, at 50% sparsity, SDS outperforms SparseGPT by approximately 0.5% and 0.8% for OPT-1.3b and OPT-2.7b, respectively. In the 2:4 sparsity configuration, SDS marginally lags behind SparseGPT for OPT-2.7b; this phenomenon may be attributed to the high sensitivity of perplexity as an evaluation metric. Importantly, zero-shot evaluations on downstream tasks (cf., Table 3) validate that SDS is not inferior to SparseGPT, solidifying SDS as a robust choice to prune PLMs.

**Zero-shot performance on downstream tasks.** Downstream tasks offer a more nuanced view, allowing us to test the model's capability across a range of different linguistic and reasoning challenges. Therefore, we conducted an empirical study to compare the performance of different sparse configurations of the pruned OPTs on a wide range of downstream tasks. Our primary focus is evaluating zero-shot performance, a setup where the models are not fine-tuned and make predictions based on constant parameters. We chose seven representative downstream tasks for this evaluation: COPA, Lambada, OpenbookQA, PIQA, RTE, StoryCloze, and Winogrande. Experimental results are shown in Table 3.

Experimental results highlight the consistent superiority of the SDS framework over the SparseGPT baseline across various downstream tasks and model sizes. On average, for the 50% sparsity level,

Table 3: **Multitasking zero-shot performance.** Accuracy (%) was obtained by zero-shot evaluation and averaging over seven downstream tasks, including COPA, Lambada, OpenbookQA, PIQA, RTE, StoryCloze, and Winogrande.

| Sparsity | Method | OPT-125m | OPT-350m | OPT-1.3b | OPT-2.7b |
|---|---|---|---|---|---|
| 0 | Dense | 50.82 | 54.12 | 60.83 | 62.81 |
| 50% | SparseGPT | 48.85 | 52.33 | 55.89 | 61.14 |
| | SDS | **50.80** | **54.51** | **58.42** | **61.78** |
| 2:4 | SparseGPT | 47.56 | 48.34 | 53.57 | 58.48 |
| | SDS | **49.61** | **50.50** | **56.67** | **59.96** |
| 4:8 | SparseGPT | 48.29 | 49.85 | 54.95 | 60.24 |
| | SDS | **49.67** | **52.25** | **57.92** | **61.48** |

SDS surpasses SparseGPT by approximately 1.83%. This performance gain is even more pronounced at the 2:4 sparsity configuration, where SDS exceeds SparseGPT by an average of about 2.2%. At the 4:8 sparsity configuration, SDS maintains a lead, outperforming SparseGPT by an average margin of approximately 1.75%.

In summary, our evaluations convincingly demonstrate the robustness and efficacy of the SDS framework across a variety of sparsity configurations. Both language modeling and zero-shot downstream multitask performance metrics affirm the consistent superiority of SDS over the SparseGPT baseline. Therefore, SDS is an efficient and effective pruning method for PLMs.

## 3.3 ABLATION STUDY

Table 4: **Comparison of different configurations of the SDS framework.** We compare the language understanding perplexity and accuracy of OPT-125m on eight tasks in a 2:4 sparse configuration. The gray characters represent the skipped steps; DD stands for dense data, which uses the activations generated by the dense model as inputs for weight adjustment; SD stands for sparse data, which uses the activations generated by the sparse model as inputs for weight adjustment; KD stands for KD-aware data, which uses the activations of the model after weight adjustment as inputs for the next layer of weight adjustment; WR represents weight regularization; MSD stands for multiple sparse data, which means that different samples are used for each step of the SDS process.

|      | Method     | Wiki.↓ | COPA↑ | Lamb.↑ | BookQ.↑ | PIQA↑ | RTE↑  | Story.↑ | Wino.↑ | Avg(acc.)↑ |
|------|------------|--------|-------|--------|---------|-------|-------|---------|--------|------------|
| 1:   | **Dense**  | 27.66  | 66    | 39.16  | 28.0    | 62.02 | 50.18 | 60.03   | 50.36  | 50.82      |
| 2:   | **S**DS    | 60.43  | 62    | 27.55  | 25.8    | 57.24 | 53.79 | 55.38   | 51.14  | 47.56      |
| 3:   | SD**S** w DD | 58.63 | 62   | 27.03  | 26.0    | 58.11 | 52.35 | 55.25   | 50.75  | 47.36      |
| 4:   | SD**S** w SD | 58.56 | 62   | 20.96  | 26.4    | 58.65 | 51.62 | 56.21   | 50.98  | 46.69      |
| 5:   | SD**S** w KD | 56.82 | 63   | 30.04  | 26.2    | 58.76 | 53.79 | 55.63   | 49.64  | 48.15      |
| 6:   | S**DS**    | 57.98  | 62    | 26.68  | 26.2    | 59.30 | 48.74 | 56.27   | 50.98  | 47.17      |
| 7:   | **SDS** w/o WR | 51.96 | 63 | 30.04  | 26.2    | 58.92 | 51.95 | 56.46   | 51.38  | 48.29      |
| 8:   | **SDS** w DD | 57.72 | 62   | 26.99  | 26.0    | 59.30 | **54.15** | 55.19 | **51.46** | 47.87   |
| 9:   | **SDS** w KD | 57.32 | 61   | 29.15  | 26.4    | 59.51 | 54.01 | 54.17   | 51.38  | 47.95      |
| 10:  | **SDS** w SD | **51.30** | **65** | **31.57** | **27.8** | 59.85 | **54.15** | **57.42** | 51.46 | **49.61** |
| 11:  | **SDS** w MSD | 52.06 | 64  | 30.35  | 27.0    | **60.01** | 50.18 | 57.16 | **52.57** | 48.75    |

To validate the effectiveness of *the step composition* and *the sparse regularization* of the Sparse-Dense-Sparse (SDS) framework, we conducted a series of ablation experiments as shown in Table 4. The first two rows represent the dense and sparse baseline, respectively.

Rows 3 to 5 verify the effect of only performing once pruning and sparse weight adjustment. Overall, SD**S** was only able to outperform SparseGPT on three tasks completely. Comparing the different input data used in the SD**S** case, SD**S** w KD can outperform SparseGPT on seven tasks, which is considered better than choosing the other two data types. Thus it can be concluded that *SDS mode has limited optimization for SparseGPT* and *selection of data with low loss (KD) is more suitable for SDS mode than selection of data with high loss (DD or SD)*.

Row 6 verifies the effect of the second round pruning of the dense model after injecting it directly with sparse regularization, skipping the initial pruning, i.e. sparse inherited traits. S**DS** outperforms the performance of SparseGPT on five tasks, but it does not yet reach the superior performance of **SDS** w SD. This observation demonstrates that *sparse inherited traits are effective*.

Rows 7 to 10 verify the role of weight-based and data-based regularization in **SDS**, respectively. Unlike SD**S**, SD is a more suitable data choice for **SDS**, and this harder data serves the purpose of regularization while avoiding the challenge of learning hard data in multiple steps. Also, it can be argued that sparse inherited traits and data regularization dominate in sparse regularization compared to weight regularization. A.3 provides an analysis from a distributional perspective.

Row 11 shows the impact of using different samples at each step of the SDS process. The optimization is closer to SDS w SD, but only two tasks outperform it. In the short term, the addition of more

samples does not improve the performance of SDS, which may be caused by the failure to learn sufficiently about each batch of samples as well as the varying quality of the samples, but confirms that SDS does not need to rely on additional samples. The effect of introducing more samples and implementing SDS optimization in iterations can be referenced in Section A.4.

# 4 RELATED WORKS

**Pruning for language model compression**. The surging complexity of Transformer-based language models, which now feature up to hundreds of billions of parameters, has accentuated the urgent need for effective and efficient model pruning methods (Han et al., 2016; 2015; Hassibi et al., 1993; Mishra et al., 2021; Pool & Yu, 2021b; Ma et al., 2023; Liu et al., 2023). These pruning methods can be broadly classified into structured and unstructured approaches. Structured pruning is more hardware-friendly, as it directly prunes entire segments of weights, thereby removing consecutive computations. As an example, LLM-Pruner (Ma et al., 2023) has been proposed for structured pruning of PLMs, which employs a gradient-based approach to selectively remove non-critical structure groups, aiming to reduce the model size while retaining the performance of PLMs. In contrast to the static pruning used in LLM-Pruner, DejaVu (Liu et al., 2023) introduces a dynamic pruning method that prunes different positions of the model based on contextual input information. Additionally, unstructured pruning is also receiving interest, particularly as hardware advancements increasingly support the acceleration of sparse patterns such as 2:4 or 4:8 sparse (Mishra et al., 2021). Techniques such as SparseGPT (Frantar & Alistarh, 2023) extend the OBS (Hassibi et al., 1993) methodology to prune weights column by column, allowing the modification of values in the unpruned columns to compensate for the pruning errors. Syed et al. (2023) enhances SparseGPT by incorporating minimal iterative task fine-tuning during the pruning process, demonstrating performance improvements at high sparsity levels. Wanda (Sun et al., 2023) introduces a simple yet effective no-retraining-needed pruning strategy that prunes weights based on their magnitudes and corresponding activations. LoSparse (Li et al., 2023) enhances model compression by approximating a weights matrix as the sum of a low-rank matrix and a sparse matrix (Hu et al., 2022), synergizing the benefits of both low-rank approximations and pruning techniques.

**Weight distribution optimization**. Various techniques have been employed to understand and optimize weight distributions in the quest for more efficient neural networks. The Dense-Sparse-Dense training method (Han et al., 2017) provides a three-step flow: an initial dense training to learn connection weights, a sparsity-inducing phase that prunes unimportant connections, and a final re-dense step. This process improves performance across various network architectures and underscores the importance of parameter distribution in achieving better local optima. Regularization methods serve as pivotal tools for optimizing the parameter distribution. Dropout (Srivastava et al., 2014) is a form of ensemble learning to the neural networks. It implicitly changes the parameter distribution by randomly zeroing out weights during training, encouraging a sparse representation. Yoshida & Miyato (2017) focuses on constraining the spectral norm of weights matrices to improve the generalization capabilities of neural networks. This method plays a crucial role in shaping the parameter space, making it more amenable to sparse approximations.

In this paper, the proposed Sparse-Dense-Sparse (SDS) framework first regularizes the weights into a pruning-friendly dense distribution and prunes the models, aiming to enhance the language comprehension and multitasking performance of the state-of-the-art pruning method SparseGPT.

# 5 CONCLUSION

We introduced the Sparse-Dense-Sparse (SDS) framework for optimizing pruned generative pre-trained language models (PLMs), consisting of initial pruning, re-dense weight reconstruction, and a second pruning round. The SDS framework focuses on weight distribution optimization and incorporates sparse regularization elements—including inherited traits, data-based regularization, and weight-based regularization. As a result, SDS not only enhances the model's pruning friendliness but also achieves state-of-the-art pruning results. Experimental results show that SDS surpasses SparseGPT and Wanda by reducing language comprehension perplexity by an average of $6.4$ and increasing the overall accuracy by $1.8\%$ across seven downstream tasks on OPT, GPT, and LLaMA. The SDS framework presents a different choice for the effective and efficient pruning of PLMs.

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

# A APPENDIX

## A.1 THE SPARSE-DENSE-SPARSE FRAMEWORK

---

**Algorithm 1** The sparse-dense-sparse (SDS) framework

---

**Input:** Pre-trained dense model $\mathbb{W}^{\text{dense}} = \{\boldsymbol{W}_1^{\text{dense}}, \boldsymbol{W}_2^{\text{dense}}, ..., \boldsymbol{W}_l^{\text{dense}}\}$
**Output:** Final pruned model $\widehat{\mathbb{W}}^{\text{SDS}} = \{\widehat{\boldsymbol{W}}_1^{\text{SDS}}, \widehat{\boldsymbol{W}}_2^{\text{SDS}}, ..., \widehat{\boldsymbol{W}}_l^{\text{SDS}}\}$

———————————————————— **Initial pruning** ————————————————————

**Require:** Original unlabeled samples $\boldsymbol{X}$; sparsity
1: **for each** $\boldsymbol{W}^{\text{dense}}$ **in** $\mathbb{W}^{\text{dense}}$ **do**
2:      $\boldsymbol{W}^{\text{sparse}} = \text{empty}(\boldsymbol{W}^{\text{dense}})$
3:      $\boldsymbol{H} = \boldsymbol{X}\boldsymbol{X}^{\top}$
4:      $s = \text{sort}\left(\frac{\boldsymbol{W}^{\text{dense}\,2}}{\text{diag}(\boldsymbol{H}^{-1})^2}\right)$
5:      $\boldsymbol{M} = \mathbb{1}\left(s > \text{sparsity}\right)$
6:      **for** $c = 1$ to $\text{column\_size}(\boldsymbol{W}^{\text{dense}})$ **do**
7:          $\boldsymbol{W}_{:,c}^{\text{sparse}} = \boldsymbol{M}_{:,c} \odot \boldsymbol{W}_{:,c}^{\text{dense}}$                $\triangleright$ Prune one column with mask $\boldsymbol{M}$
8:          $\boldsymbol{W}_{:,c+1:}^{\text{dense}} = \boldsymbol{W}_{:,c+1:}^{\text{dense}} - \frac{\left(\boldsymbol{W}_{:,c}^{\text{sparse}} - \boldsymbol{W}_{:,c}^{\text{dense}}\right)^2}{[\boldsymbol{H}^{-1}]_{c,c}^2} \cdot \boldsymbol{H}_{:,c}^{-1}$                $\triangleright$ Error compensation
9:      **end for**
10:     $\boldsymbol{X} \leftarrow \boldsymbol{W}^{\text{sparse}}\boldsymbol{X}$                $\triangleright$ Error accumulation
11: **end for**

———————————————— **Re-dense weight reconstruction** ————————————————

**Require:** Pre-trained dense model $\mathbb{W}^{\text{dense}} = \{\boldsymbol{W}_1^{\text{dense}}, \boldsymbol{W}_2^{\text{dense}}, ..., \boldsymbol{W}_l^{\text{dense}}\}$;
          Initial pruned sparse model $\mathbb{W}^{\text{sparse}} = \{\boldsymbol{W}_1^{\text{sparse}}, \boldsymbol{W}_2^{\text{sparse}}, ..., \boldsymbol{W}_l^{\text{sparse}}\}$;
          Original unlabeled samples $\boldsymbol{X}$; learning rate $\eta$;
          L1 regularization ratio $\lambda_1$; L2 regularization ratio $\lambda_2$
1: **for** $i = 1$ **to** $l$ **do**
2:      **while** not converged **do**
3:          $\boldsymbol{W}_i^{\text{re-dense}(t)} = \boldsymbol{W}_i^{\text{re-dense}(t-1)} - \eta\nabla\left\|\boldsymbol{W}_i^{\text{dense}}\boldsymbol{X} - \boldsymbol{W}_i^{\text{re-dense}(t-1)}\boldsymbol{X}\right\|_2^2$

             $-\eta\lambda_1\nabla\left\|\boldsymbol{W}_i^{\text{re-dense}(t-1)}\right\|_1 - \eta\lambda_2\nabla\left\|\boldsymbol{W}_i^{\text{re-dense}(t-1)}\right\|_2^2$
4:      **end while**
5:      $\boldsymbol{X} \leftarrow \boldsymbol{W}_i^{\text{sparse}}\boldsymbol{X}$                $\triangleright$ Error accumulation
6: **end for**

———————————— **Second pruning: sparse weight adjustment** ————————————

**Require:** Pre-trained dense model $\mathbb{W}^{\text{dense}} = \{\boldsymbol{W}_1^{\text{dense}}, \boldsymbol{W}_2^{\text{dense}}, ..., \boldsymbol{W}_l^{\text{dense}}\}$;
          Re-dense trained model $\mathbb{W}^{\text{re-dense}} = \{\boldsymbol{W}_1^{\text{re-dense}}, \boldsymbol{W}_2^{\text{re-dense}}, ..., \boldsymbol{W}_l^{\text{re-dense}}\}$;
          Original unlabeled samples $\boldsymbol{X}$; learning rate $\eta$; sparsity
1: Repeat the pruning process and yield $\mathbb{W}^{\text{sparse-2nd}} = \{\boldsymbol{W}_1^{\text{sparse-2nd}}, \boldsymbol{W}_2^{\text{sparse-2nd}}, ..., \boldsymbol{W}_\ell^{\text{sparse-2nd}}\}$
2: **for** $i = 1$ **to** $l$ **do**
3:      **while** not converged **do**
4:          $s = \text{sort}\left(\left|\boldsymbol{W}_i^{\text{SDS}(t)}\right|\right)$
5:          $\boldsymbol{M} = \mathbb{1}\left(s > \text{sparsity}\right)$
6:          $\boldsymbol{W}_i^{\text{SDS}(t)} = \boldsymbol{M} \odot \left(\boldsymbol{W}_i^{\text{SDS}(t-1)} - \eta\nabla\left\|\boldsymbol{W}_i^{\text{dense}}\boldsymbol{X} - \boldsymbol{W}_i^{\text{SDS}(t-1)}\boldsymbol{X}\right\|_2^2\right)$
7:      **end while**
8:      $\boldsymbol{X} \leftarrow \boldsymbol{W}_i^{\text{sparse-2nd}}\boldsymbol{X}$                $\triangleright$ Error accumulation
9: **end for**

---

## A.2 ERROR ACCUMULATION AND DATA REGULARIZATION

The input data used in weight adjustment can be categorized in two ways: whether to perform error accumulation and whether to be aware of the knowledge distillation (KD) process. Figure 5 presents four different data selection ways for weight adjustment.

**(a)** Weight adjustment with KD aware and error accumulation, this paradigm corresponds to *KD-data* in our ablation study (cf., section 3.3): after applying KD to the sparse layer, a subsequent forward propagation is needed to generate inputs for the next layer. These inputs are solely based on the former layer's

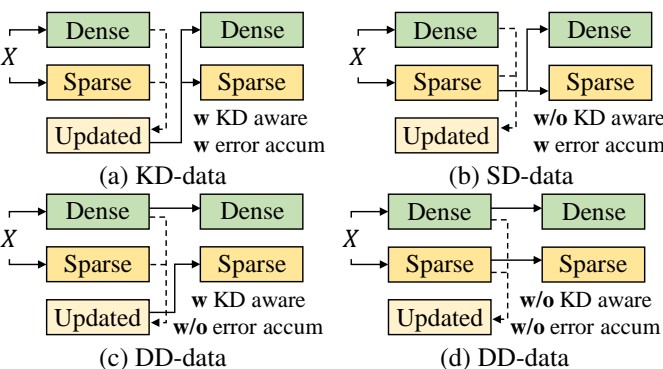

Figure 5: **Four data selection paradigms in weight adjustment.** Straight lines represent forward propagation and dashed lines represent knowledge distillation.

outputs, thus accumulating errors. Since KD aims to reduce loss, this extra forward propagation simplifies the data, making it easier for the subsequent layer to learn. **(b)** Weight adjustment with error accumulation but without KD aware, this paradigm corresponds to *SD-data* in our ablation study: unlike paradigm (a), this approach abandons the additional forward propagation to account for changes in the layer updated by KD. This results in the next layer of learning from data corresponding to a higher loss, making learning more challenging than in paradigm (a). **(c)** and **(d)** are two ways of adjusting the weights without accumulating errors. The presence or absence of KD awareness has a minimal impact on either, as the optimization direction is constrained by the same dense model in both cases. The *DD-data* paradigm in our ablation study employs paradigm (d).

From the perspective of data difficulty, *DD-data* is the most difficult because it requires each layer to compensate for the errors accumulated in all previous layers. This difficulty is more prominent in the KD process under sparsity constraints. In the ablation study (cf., section 3.3), neither one nor two times optimization of the sparse model using *DD-data* was able to achieve excellent results, verifying the above observation. *KD-data* is the easiest because the weights of the sparse model are updated in the direction of lower loss during knowledge distillation. The use of *KD-data* has yielded relatively good results only in single-step optimization of the sparse model due to the fact that simple data carries less data regularization and a relatively low upper bound for optimization. *SD-data* is relatively moderate in difficulty and comes with data regularization and hence achieved an ideal result in SDS's optimization of the sparse model. The reason why *SD-data* did not achieve an ideal result in the single-step optimization could be the challenge of the difficult data.

Not only the weight adjustment but also the pruning process faces the issue of data selection; in the absence of weight adjustment, the only data available for the pruning process are *DD-data* and *SD-data* (*KD-data* degenerates into *SD-data*). *DD-data* can be considered as ideal data, and *SD-data* can be considered as real data (the ab-

Table 5: **Effect of data selection on pruning.** The language modeling perplexity of the OPT-125m model was evaluated on raw-WikiText2 using a calibration set sourced from C4.

| Method | 50% sparse | 2:4 sparse | 4:8 sparse |
|---|---|---|---|
| SparseGPT w DD | 37.32 | 61.81 | 44.77 |
| SparseGPT w SD | 36.85 | 60.43 | 44.77 |

sence of knowledge distillation makes the data difficulty perspective less appropriate). The effect of SparseGPT pruning using different data is shown in Table 5.

According to the experimental results, the availability of both data is guaranteed. Besides, based on the fact that *SD-data* is better than *DD-data*, it is possible to conclude that real data is more appropriate than ideal data for calibration in pruning.

## A.3 DISTRIBUTION ANALYSIS

Figure 6 visualizes the impact of several pertinent optimizations performed on pruned PLMs from the perspective of distribution changes.

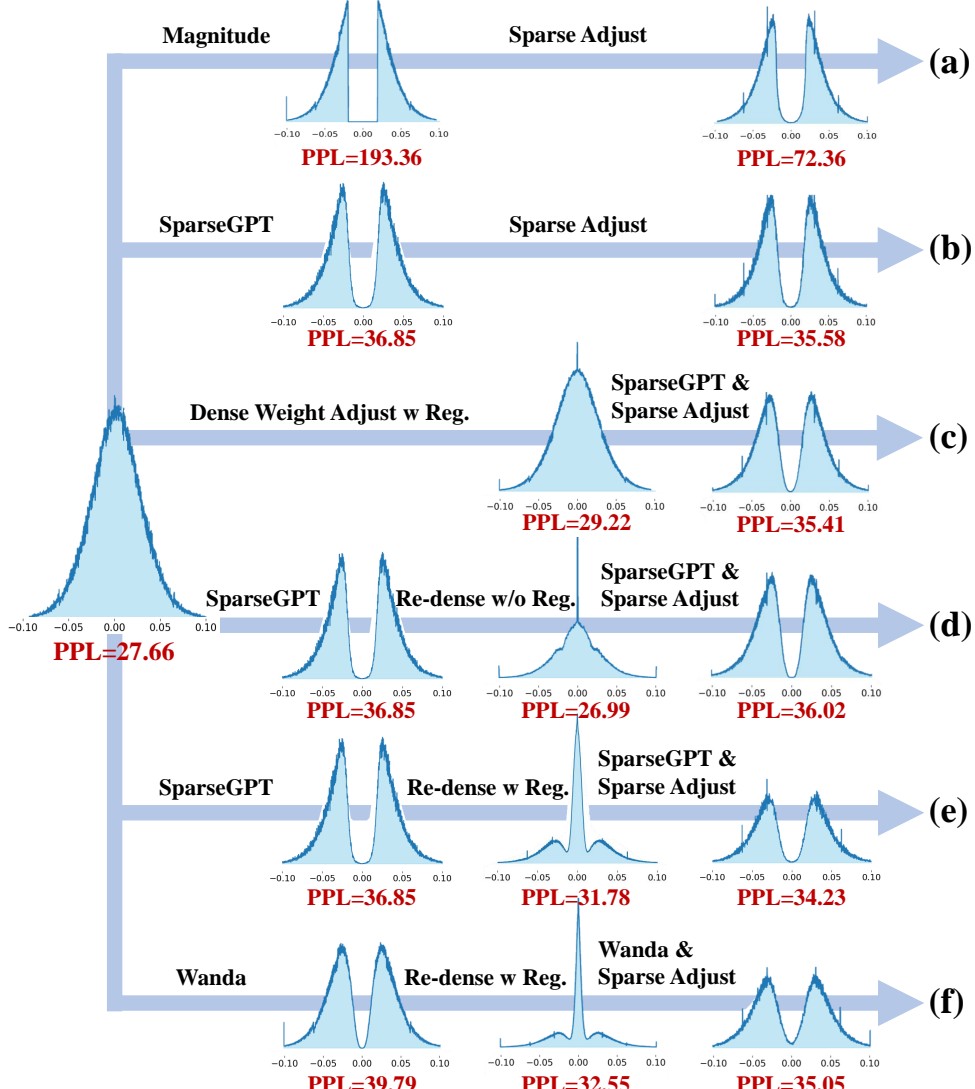

Figure 6: **Changes in distributions during optimization of pruned PLMs.** The distribution observations are from the last layer of OPT-125m with 50% pruning. **(a)** represents the process of first pruning the model by magnitude (*absmin*) (Hagiwara, 1994) and then optimizing the pruned model using SD-data. **(b)** represents the SDS w KD in the ablation study (cf., section 3.3). **(c)** represents the SDS. **(d)** represents the **SDS** w KD. **(e)** represents the **SDS** w SD. **(f)** represents the **SDS** w SD and with Wanda as the pruning method. Zero values are omitted in sparse weight distributions for better clarity.

Magnitude-based one-shot pruning method is ineffective on PLMs primarily because it focuses only on the absolute value of the weights. This simplistic approach tends to create a truncated bimodal distribution of the model weights, concentrating them at extreme positive and negative values. Distribution truncation can lead to model instability, as removing near-zero weights disrupts the model's ability to make subtle, nuanced adjustments. Due to the large amount of information lost in the pruning process, the model's performance can only be recovered to a limited extent after weight adjustment. In contrast, modern pruning methods like SparseGPT take into account higher-order

rather than zero-order information, which manages to maintain an untruncated bimodal distribution similar to what magnitude pruning plus subsequent weight adjustment would achieve. However, they do it in a single step and are able to achieve better performance.

As shown in Figure 6b, the model has a relatively sharp bimodal peak in its distribution after being pruned by SparseGPT, which challenges the model's generalization ability, optimization space and stability. Direct adjustment of the pruned model's weights yields limited performance and optimization of the weight distribution. Therefore, it is necessary to consider the SDS process.

Before attempting the SDS process, Figures 6c and 6d show the trend of weight distribution changes for only injecting regular regularization or sparse inherited traits into the model, respectively. Both find a new dense solution to some extent: a dense model with a smoother distribution and more zeros can be found by using data-based regularization and weight-based regularization for dense weight adjustment, and a dense model that converges to a multi-peaked distribution with more zeros is obtained after re-dense reconstruction of the sparse model without regular regularization. After a second round of pruning, both approaches lead to a recovery in the model's performance. However, they are not as effective as the SDS process that uses a combination of data-based regularization, weight-based regularization, and sparse inherited traits, as shown in Figure 6e and Figure 6f. This illustrates the effectiveness and mutual reinforcement effect of regularization techniques in the SDS framework. An interesting phenomenon is that when regular regularization is not used, it is possible to reconstruct the pruned model to equal or even higher performance than the original dense model. This is perhaps due to the absence of regularization techniques, which allowed the re-dense model to overfit the behavior of the original dense model. The limited performance improvement of the re-dense model after a second round of pruning also supports the above deduction.

## A.4 ITERATIVELY IMPLEMENTING SPARSE-DENSE-SPARSE OPTIMIZATION

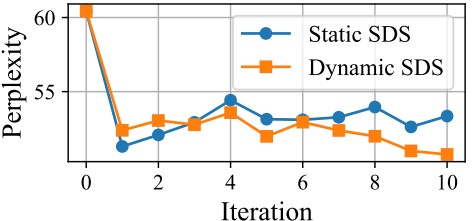 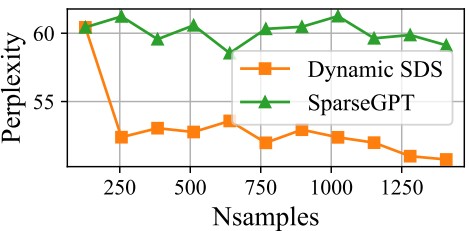

Figure 7: **The impact of iteratively applying sparse-dense-sparse optimization, specifically contrasting the outcomes when employing either identical samples (Static SDS) or varied samples (Dynamic SDS) throughout the process.** OPT-125m is utilized in this experiment with a pruning configuration of 2:4 sparsification. The left figure illustrates a comparison between static and dynamic SDS results across multiple iterations. The right figure demonstrates a comparison between SparseGPT and SDS, depicting variations in performance as the size of the calibration set changes.

Grounded in SDS, we would like to verify whether iteratively applying the sparse-dense-sparse optimization can further improve the performance of the model. On the one hand, we examine the impact on model performance of using exactly the same samples as a calibration set in the iterations (Static SDS) so that the number of samples used in the optimization remains 128. On the other hand, we examine the effect of using different samples in each iteration on the model's performance (Dynamic SDS), i.e., at iteration $i$, the optimization uses a sample size of $i \times 128$.

As shown left in Figure 7, implementing SDS once (Iteration 1) compared to SparseGPT (Iteration 0) can bring huge improvements, whether using the same or different calibration samples. However, as the number of iterations increased, implementing the SDS optimization multiple times with the same calibration samples failed to improve the model's performance. This is because the model that is optimized excessively on the same batch of samples is prone to overfitting, leading to performance degradation. When different samples were used, superior results to a single implementation of SDS were produced. This demonstrates that data diversity can positively affect the iterative SDS optimization process. However, the not strictly monotonically increasing performance with iteration also seems to emphasize that the inclusion of high-quality samples may have resulted in substantial contributions.

Table 6: **Perplexity on raw-WikiText2.** SparseGPT and Wanda form the components of the SDS framework, represented as SDS-SparseGPT and SDS-Wanda, respectively. The calibration set utilized is C4.

| Sparsity | Method | OPT-125m | OPT-350m | OPT-1.3b | OPT-2.7b | GPT2-s | GPT2-m | GPT2-l | GPT2-xl |
|---|---|---|---|---|---|---|---|---|---|
| 0 | Dense | 27.66 | 22.01 | 14.62 | 12.16 | 29.95 | 21.71 | 19.33 | 17.41 |
| 50% | SparseGPT | 36.85 | 31.58 | 17.46 | 13.48 | 47.46 | 28.06 | 23.55 | 19.91 |
| | SDS-SparseGPT | **34.23** | **29.36** | **17.40** | **13.42** | **45.76** | **27.91** | **23.32** | **19.62** |
| | Wanda | 39.79 | 41.88 | 18.36 | 14.38 | 46.71 | 29.29 | 24.89 | 20.83 |
| | SDS-Wanda | **35.05** | **33.07** | **17.23** | **13.74** | **40.32** | **27.39** | **23.15** | **19.78** |
| 2:4 | SparseGPT | 60.43 | 51.11 | 23.90 | 17.18 | 73.11 | 40.41 | 32.49 | 25.97 |
| | SDS-SparseGPT | **51.30** | **46.23** | **23.02** | 17.36 | **64.31** | **38.24** | **31.33** | **25.05** |
| | Wanda | 82.47 | 113.17 | 27.32 | 20.94 | 123.66 | 61.70 | 52.39 | 32.60 |
| | SDS-Wanda | **59.17** | **73.56** | **23.94** | **18.02** | **63.57** | **41.11** | **31.11** | **25.35** |
| 4:8 | SparseGPT | 44.77 | 39.59 | 19.95 | 14.98 | 53.14 | 32.84 | 26.77 | 22.70 |
| | SDS-SparseGPT | **41.66** | **34.18** | **19.54** | **14.81** | **50.90** | **32.41** | **26.29** | **22.27** |
| | Wanda | 53.97 | 62.49 | 21.96 | 16.80 | 73.73 | 41.12 | 32.58 | 25.14 |
| | SDS-Wanda | **43.58** | **47.31** | **19.82** | **15.45** | **51.05** | **33.55** | **26.16** | **22.11** |

The right side of Figure 7 demonstrates that more samples can bring more substantial performance gains to the SDS iterative optimization process compared to SparseGPT. While using different samples can result in a better-performing model, these samples still come from the same distribution, and there is still a risk of overfitting by having the model optimized with them as the calibration set many times. Therefore, optimizing with a small number of unlabeled samples less frequently remains our primary intention, and iteratively implementing sparse-dense-sparse optimization is an optional approach that we do not recommend.

## A.5 SDS WORKS WELL ON OTHER MODELS AND PRUNING METHODS

**The general applicability of SDS in other undersized PLMs and Wanda pruning method.** To verify the general applicability of the SDS framework, we additionally chose to perform experimental validation on the GPT2 (Radford et al., 2019) models and the Wanda pruning method.

Among them, the GPT2 model covers four versions of model instances, including small (s), medium (m), large (l), and xlarge (xl) ones, with parameter sizes ranging from 124M to 1.5B approximately.

The Wanda pruning method considers both weights and activations as a saliency metric $\mathbf{S}$ for finding an efficient sparse mask:

$$\mathbf{S}_{ij} = |\mathbf{W}_{ij}| \cdot \|\mathbf{X}_j\|_2 , \tag{8}$$

where $|\cdot|$ denotes the absolute value operator, and $\|\mathbf{X}_j\|_2$ computes the $l_2$ norm of the $j$th features gathered across token dimension. The final saliency score is ascertained by the multiplication of these two scalar values. Compared to SparseGPT, Wanda is able to achieve large model pruning similar to SparseGPT without weight modification, which contributes to the simplicity of Wanda.

Tables 6 and 7 show the results of the complete language modeling perplexity and downstream multi-task zero-shot experiments, respectively.

Based on the results of the empirical evaluation, it is evident that the Sparse-Dense-Sparse (SDS) approach significantly outperforms both SparseGPT and Wanda methods across multiple sparsity configurations and model sizes. In terms of language modeling perplexity, SDS demonstrates a clear advantage, improving on average by around 11% over SparseGPT across various sparsity configurations and model sizes. It also shows significant improvements over Wanda, with an average gain of roughly 18% in perplexity metrics. In the multitask zero-shot experiment, the superiority of SDS is further evidenced; it outperforms SparseGPT by an average of 2.5% and surpasses Wanda by an average of 3.1%.

Table 7: **Multitasking zero-shot accuracy comparison.** Accuracy (%) is obtained by zero-shot evaluation and averaging over seven downstream tasks, including COPA, Lambada, OpenbookQA, PIQA, RTE, StoryCloze, and Winogrande.

| Sparsity | Methpd | OPT-125m | OPT-350m | OPT-1.3b | OPT-2.7b | GPT2-s | GPT2-m | GPT2-l | GPT2-xl |
|---|---|---|---|---|---|---|---|---|---|
| 0 | Dense | 50.82 | 54.12 | 60.83 | 62.81 | 50.07 | 53.58 | 56.51 | 58.40 |
| 50% | SparseGPT | 48.85 | 52.33 | 55.89 | 61.14 | 47.27 | 52.82 | 53.47 | 57.22 |
| | SDS-SparseGPT | **50.80** | **54.51** | **58.42** | **61.78** | **48.37** | **53.33** | **54.34** | **58.00** |
| | Wanda | 48.46 | 48.90 | 56.18 | 59.36 | 46.50 | 52.01 | 53.65 | 56.03 |
| | SDS-Wanda | **49.78** | **51.40** | **57.58** | **60.92** | **49.05** | **53.34** | **54.88** | **57.36** |
| 2:4 | SparseGPT | 47.56 | 48.34 | 53.57 | 58.48 | 46.47 | 50.17 | 50.85 | 53.57 |
| | SDS-SparseGPT | **49.59** | **50.50** | **56.67** | **59.96** | **47.62** | **50.65** | **52.45** | **56.25** |
| | Wanda | 45.69 | 44.77 | 52.86 | 55.51 | 42.32 | 47.38 | 48.92 | 51.37 |
| | SDS-Wanda | **47.09** | **46.69** | **54.44** | **58.98** | **46.55** | **51.48** | **52.77** | **54.68** |
| 4:8 | SparseGPT | 48.29 | 49.85 | 54.94 | 60.24 | 46.32 | 51.04 | 52.53 | 55.77 |
| | SDS-SparseGPT | **49.67** | **52.25** | **57.92** | **61.48** | **48.00** | **52.15** | **53.45** | **55.85** |
| | Wanda | 46.28 | 46.41 | 55.04 | 58.21 | 44.63 | 49.15 | 51.17 | 54.18 |
| | SDS-Wanda | **47.70** | **48.61** | **56.12** | **59.91** | **47.07** | **52.04** | **54.44** | **56.90** |

**The general applicability of SDS in larger and stronger PLMs.** The LLaMA model is considered to be more adequately trained and harder to compress (Touvron et al., 2023a;b), and we choose the LLaMA model to further validate the effectiveness of the SDS framework. Specifically, we take the 7b-sized models of LLaMA and LLaMA2 with 2:4 sparse as the pruning configuration and SparseGPT/Wanda as the base pruning method to verify the language modeling perplexity and multi-task performance of the SDS-optimized pruned models, as shown in Table 9.

Analysis of SDS methods in LLaMA-7b and LLaMA2-7b models shows an average perplexity reduction of approximately 10.90% and an accuracy increase of around 2.46%. These findings demonstrate the SDS's effectiveness in enhancing larger and stronger PLMs' precision and efficiency.

In summary, these findings confirm the superiority of the SDS framework. SDS shows strong performance enhancement effects either for undersized or larger pruned models, making it an efficient and effective strategy for model pruning and performance optimization.

**Execution efficiency of the SDS framework.** Table 8 demonstrates the time required to perform SDS optimization. SDS employs a layer-by-layer knowledge distillation strategy, allowing for parallel processing of individual layer optimizations. The time consumption for layer-serial SDS optimization using a single 32GB Nvidia V100 GPU ranges from 22 minutes to 34 hours as the model grows larger.

Table 8: **Time consuming of SDS optimization.** There are time consumption of single-device serial optimization, multi-device parallel optimization, and theoretical ultimate parallel optimization as the weight adjustment type.

| Type | 125M | 1.3B | 2.7B | 7B |
|---|---|---|---|---|
| Serial | $\sim$ 22 min. | $\sim$ 3.8 hours | $\sim$ 7.4 hours | $\sim$ 34.2 hours |
| Parallel | $\sim$ 4 min. | $\sim$ 25 min. | $\sim$ 1 hours | $\sim$ 4.5 hours |
| Theoretical | <1 min. | $\sim$ 6 min. | $\sim$ 8 min. | $\sim$ 39 min. |

Since SDS optimization uses *SD-data* (cf., Section A.2), which is available in advance, and SDS does not need to perform an additional forward propagation for each optimized layer, each layer in the network can perform optimization **in parallel**. On eight 32GB Nvidia V100 GPUs, we can optimize layers within individual GPUs serially and layers between GPUs in parallel, so it only took us from 4 minutes to 4.5 hours to perform the SDS optimization. With GPU device redundancy or GPU memory redundancy, the granularity of parallelism can be as low as a single layer, thus yielding the theoretically fastest optimization time consumption, ranging from 1 minute to 39 minutes. In summary, the process of running SDS is efficient.

Table 9: **2:4 Pruned LLaMA performance.** SparseGPT and Wanda form the components of the SDS framework, represented as SDS-SparseGPT and SDS-Wanda, respectively. The calibration set utilized is C4.

| PLM | Method | Wiki.↓ | COPA↑ | BookQ.↑ | PIQA↑ | RTE↑ | Story.↑ | Wino.↑ | Avg(acc.)↑ |
|---|---|---|---|---|---|---|---|---|---|
| | Dense | 5.68 | 84 | 42.4 | 77.48 | 53.43 | 75.94 | 67.01 | 66.71 |
| | SparseGPT | 11.23 | **80** | 35.4 | 70.62 | 59.57 | 69.89 | 59.98 | 62.58 |
| LLaMA-7b | SDS-SparseGPT | **9.97** | 79 | **35.6** | **71.65** | **59.57** | **71.67** | **61.80** | **63.22** |
| | Wanda | 11.54 | 80 | 36.4 | 68.88 | **58.12** | 69.00 | 60.06 | 62.08 |
| | SDS-Wanda | **9.73** | **82** | **36.8** | **70.57** | 57.60 | **70.27** | **62.04** | **63.21** |
| | Dense | 5.47 | 87 | 40.8 | 77.04 | 61.73 | 77.59 | 67.01 | 68.53 |
| | SparseGPT | 10.82 | 83 | 35.6 | 71.33 | 63.90 | 71.23 | 63.30 | 64.73 |
| LLaMA2-7b | SDS-SparseGPT | **10.02** | **84** | **37.2** | **71.44** | **64.09** | **71.99** | **63.88** | **65.43** |
| | Wanda | 12.14 | 79 | 34.6 | 70.08 | 59.57 | 69.70 | 60.22 | 62.20 |
| | SDS-Wanda | **10.39** | **81** | **36.8** | **70.62** | **64.25** | **71.8** | **63.06** | **64.59** |

Even so, SDS optimization is still more time-consuming than SparseGPT or Wanda. However, the optimization of pruned PLMs focuses more on the performance and efficiency after the optimization is completed and the complexity of the optimization process itself is often tolerable.

## A.6 A DISCUSSION IN WEIGHT INITIALIZATION

In this paper, we enhance the performance of the pruned model by injecting sparse regularization to provide pruning-aware capabilities. Injecting awareness of subsequent operations into the model during pre-training or fine-tuning or reconstruction process, e.g., awareness of pruning, quantization, low-rank decomposition, etc., is considered efficacious.

Overall, the awareness of subsequent operations will be reflected in the weight initialization of the model. Some techniques that may be used to optimize weight initialization are as follows.

**Channel permutation.** When two neighboring weights exist, the channels of the two can be re-arranged without changing the inputs and outputs (Pool & Yu, 2021a). Specifically, the channels in the inner product direction of the first weight can be permuted arbitrarily. Computational equivalence is guaranteed if the non-inner-product directions of the second weight are permuted in a manner consistent with the former. Channel permutation has the potential to distribute important and non-important weights more evenly, avoiding situations where all $m$ weights are important or all $m$ weights are unimportant in an $n{:}m$ sparsity configuration. Channel permutation needs to pick an ideal result among several permutation candidates obtained from the search.

**Weight equalization and bias correction.** Weight equalization can reduce the performance loss in quantization due to scale differences in layers or channels (Nagel et al., 2019). Specifically, weight equalization multiplies each channel of the current layer by a set of scale factors to compress the span of weight values, while the weights of the previous layer need to be divided by the same scale factors in order to ensure computational equivalence. The bias corresponding to the weights can also be corrected to improve the model's performance, and related methods include scale smoothing, backpropagation, and so on. In the context of the need to protect outliers for quantizing PLMs, the above techniques have led to advanced quantization techniques such as SmoothQuant (Xiao et al., 2023) and AWQ (Lin et al., 2023).

The practical deployment of the model cannot be separated from the organic combination of various model compression techniques, and we hope that the idea of weight initialization can bring more model optimization inspirations for academia and industry.

