# OpenReview forum: "Enhancing One-Shot Pruned Generative Pre-training Language Models through Sparse-Dense-Sparse Mechanism"
_ICLR.cc/2024/Conference — Submitted to ICLR 2024_

### Official Review · Reviewer_XgFT · 2023-10-31

**Soundness:** 3 good
**Presentation:** 4 excellent
**Contribution:** 3 good
**Rating:** 6
**Confidence:** 4

**Summary:**

This work proposes a three-step framework to improve one-shot pruning for large language models to potentially accelerate inference. In the first step, the method performs a standard one-shot pruning such as the SparseGPT method; in the second step, they perform a dense reconstruction of the sparse model to reactivate the pruned connections, aiming to identify a dense model with enhanced pruning awareness; and in the last step, they perform pruning again for the reconstructed model. Comparison to SparseGPT methods show that this three-step method performs better than one-shot pruning with a single pruning step.

**Strengths:**

The explanation provided for Figure 3 is compelling and effectively illustrates the effect of the method on model parameters. Nevertheless, it's worth noting that the visualization in Figure 3 visualizes a small model opt-125m, leaving some uncertainty regarding whether the observed effect would hold the same significance for larger models.

**Weaknesses:**

- **Comparing to stronger methods like wandb:** Wandb (Sun et al., 2023) is a method that performs better than SparseGPT on one-shot pruning, and the authors should introduce it as a baseline and add more discussions around it. I believe that the framework is independent from the base pruning method, thus doing experiments on top of a strong existing method is highly recommended.
- **Performing experiments on stronger base models:** A clear trend that is disclosed by comparing SparseGPT and Wanda in Sun et al., 2023 is that the stronger the base model is, the more the pruning process hurts the performance of the model. For example, in the SparseGPT paper, pruning retains the model performance on OPT models; however, when Sun et al., 2023 evaluates on LLaMA based models, the performance degradation is way more significant. Such observations make intuitive sense, as the more stronger the base model is, the more information each parameter carries, and the more the model performance gets hurt when the parameter gets pruned. Given this observation, I suggest the authors test on stronger base models like LLaMA to give a more accurate account of how practical one-shot pruning is for real applications.
- **The extra step leads to diminishing returns in performance as the model scales up:** From table 2, it’s clear that as the model scales up and becomes stronger, the performance of the the re-dense and r-=prune process leads to minimal improvement compared to simply using one step of pruning.

**Questions:**

Will performing the re-dense and re-prune process in multiple iterations further improve performance?

---

> ### Author Response · Authors · 2023-11-19
>
> Dear Reviewer XgFT,
>
> We sincerely appreciate your thorough analysis and insightful comments on our manuscript, which have provided us with valuable perspectives for enhancing our manuscript. We have responded to each of your questions and provided detailed explanations and additional data to support our findings.
>
> **Response to weakness 1**: Thanks so much for your suggestion, Wanda is a very suitable one-shot pruning method as our baseline. Therefore, we construct SDS optimization experiments with Wanda as the basic pruning method.
>
> In the comparative analysis, SDS shows a significant improvement over Wanda, with an average perplexity of about 33.33, which is lower than Wanda's 42.79. In the downstream tasks, SDS also obtains an average performance improvement of 3.1% compared to Wanda. The above results illustrate that SDS enables the model performance to be improved.
>
> In fact, the Wanda method is less effective than SparseGPT for pruning undersized (non-redundant / more fully trained) PLMs. On the one hand, the idea of Wanda to protect outlier activations is less effective on undersized PLMs with insignificant outliers. On the other hand, Wanda fails to compensate for the pruning error as SparseGPT although it can find an effective sparse mask.
>
> A brief introduction to Wanda, experimental results, and more analysis can be found in **Section A.5**. *Additional comments are provided for your convenience to refer to the experimental results*.
>
> **Response for weakness 2**: As you mentioned, LLaMA is a model that is difficult to compress, and constructing experiments using LLaMA as a base model is better for exploring the validity of SDS.
>
> We take the 7b-sized models of LLaMA and LLaMA2 with 2:4 sparse as the pruning configuration and SparseGPT/Wanda as the base pruning method to verify the language modeling perplexity and multi-task performance of the SDS-optimized pruned models.
>
> Experimental results show an average perplexity reduction of approximately 10.90% and an accuracy increase of around 2.46%. These findings demonstrate the SDS's effectiveness in enhancing larger and stronger pruned PLMs' performance.
>
> For the LLaMA model, the language comprehension perplexity is impaired on a doubling scale after one-shot pruning with SparseGPT or Wanda, proving that LLaMA is indeed a PLM that is difficult to compress. LLaMA, being trained on large-scale data, is considered a non-redundant model even though it already has quite a few parameters. The fact that pruning of redundant parameters is somewhat unachievable on LLaMA elevates the difficulty of model compression and the need to perform SDS optimization.
>
> The detailed experimental results and analysis can be found in **Section A.5**. *Additional comments are provided for your convenience to refer to the experimental results*.
>
> **Response for weakness 3**: As the volume of PLM grows, the difficulty and inadequacy of its training grows rapidly. In the case of OPT-2.7b, its redundancy will be much higher than that of the smaller version OPT-125m, such that the implementation of one-shot pruning alone will already ensure very little degradation of its performance. Taking the premise that the theoretical upper bound on the performance of the SDS-optimized pruned model is on par with its dense counterpart, the effectiveness and necessity of performing SDS optimization for a highly redundant PLM is limited. However, as training techniques and high-quality data continue to be upgraded, newly born large-scale PLMs will increasingly exhibit low-redundancy characteristics, such as those of LLaMA. The SDS optimization will show more valid superiority ahead.
>
> **Response for question 1**:   Keeping the calibration set constant, and iteratively performing the sparse-dense-sparse optimization will no longer improve the performance of the pruned PLMs because of the overfitting phenomenon that arises. Better performance will result if different data is used for each iteration. Even so, we still recommend using only a single SDS to avoid potential overfitting phenomena. Specific experimental and analytical procedures can be found in **Section A.4**.
>
> **In conclusion**, we hope that our responses adequately address the concerns and suggestions you have raised. We are committed to enhancing the quality and impact of our research and believe that your feedback has significantly contributed to this goal. We look forward to any further suggestions or considerations arising from this discussion.
>
> Thank you once again for your valuable and insightful review.
>
> Best regards,
>
> Authors 5070.

---

> ### Author Response · Authors · 2023-11-22
> **Experimental results (Wikitext2 perplexity) of using Wanda as the basic pruning method**
>
> | Sparsity | Method           | OPT-125m | OPT-350m | OPT-1.3b | OPT-2.7b | GPT2-s | GPT2-m | GPT2-l | GPT2-xl |
> |----------|------------------|----------|----------|----------|----------|--------|--------|--------|---------|
> | 0        | Dense            | 27.66    | 22.01    | 14.62    | 12.16    | 29.95  | 21.71  | 19.33  | 17.41   |
> | 50%      | SparseGPT        | 36.85    | 31.58    | 17.46    | 13.48    | 47.46  | 28.06  | 23.55  | 19.91   |
> | 50%      | SDS-SparseGPT    | **34.23**| **29.36**| **17.40**| **13.42**| **45.76**| **27.91**| **23.32**| **19.62**|
> | 50%      | Wanda            | 39.79    | 41.88    | 18.36    | 14.38    | 46.71  | 29.29  | 24.89  | 20.83   |
> | 50%      | SDS-Wanda        | **35.05**| **33.07**| **17.23**| **13.74**| **40.32**| **27.39**| **23.15**| **19.78**|
> | 2:4      | SparseGPT        | 60.43    | 51.11    | 23.90    | 17.18    | 73.11  | 40.41  | 32.49  | 25.97   |
> | 2:4      | SDS-SparseGPT    | **51.30**| **46.23**| **23.02**| **17.36**| **64.31**| **38.24**| **31.33**| **25.05**|
> | 2:4      | Wanda            | 82.47    | 113.17   | 27.32    | 20.94    | 123.66 | 61.70  | 52.39  | 32.60   |
> | 2:4      | SDS-Wanda        | **59.17**| **73.56**| **23.94**| **18.02**| **63.57**| **41.11**| **31.11**| **25.35**|
> | 4:8      | SparseGPT        | 44.77    | 39.59    | 19.95    | 14.98    | 53.14  | 32.84  | 26.77  | 22.70   |
> | 4:8      | SDS-SparseGPT    | **41.66**| **34.18**| **19.54**| **14.81**| **50.90**| **32.41**| **26.29**| **22.27**|
> | 4:8      | Wanda            | 53.97    | 62.49    | 21.96    | 16.80    | 73.73  | 41.12  | 32.58  | 25.14   |
> | 4:8      | SDS-Wanda        | **43.58**| **47.31**| **19.82**| **15.45**| **51.05**| **33.55**| **26.16**| **22.11**|
>
> The above table contains the experimental results with OPT/GPT as the base model and SparseGPT/Wanda as the base pruning method.

---

> ### Author Response · Authors · 2023-11-22
> **Experimental results (average accuracy) of using Wanda as the basic pruning method**
>
> | Sparsity | Method        | OPT-125m | OPT-350m | OPT-1.3b | OPT-2.7b | GPT2-s | GPT2-m | GPT2-l | GPT2-xl |
> |----------|---------------|----------|----------|----------|----------|--------|--------|--------|---------|
> | 0        | Dense         | 50.82    | 54.12    | 60.83    | 62.81    | 50.07  | 53.58  | 56.51  | 58.40   |
> | 50%      | SparseGPT     | 48.85    | 52.33    | 55.89    | 61.14    | 47.27  | 52.82  | 53.47  | 57.22   |
> | 50%      | SDS-SparseGPT | **50.80**| **54.51**| **58.42**| **61.78**| **48.37**| **53.33**| **54.34**| **58.00**|
> | 50%      | Wanda         | 48.46    | 48.90    | 56.18    | 59.36    | 46.50  | 52.01  | 53.65  | 56.03   |
> | 50%      | SDS-Wanda     | **49.78**| **51.40**| **57.58**| **60.92**| **49.05**| **53.34**| **54.88**| **57.36**|
> | 2:4      | SparseGPT     | 47.56    | 48.34    | 53.57    | 58.48    | 46.47  | 50.17  | 50.85  | 53.57   |
> | 2:4      | SDS-SparseGPT | **49.59**| **50.50**| **56.67**| **59.96**| **47.62**| **50.65**| **52.45**| **56.25**|
> | 2:4      | Wanda         | 45.69    | 44.77    | 52.86    | 55.51    | 42.32  | 47.38  | 48.92  | 51.37   |
> | 2:4      | SDS-Wanda     | **47.09**| **46.69**| **54.44**| **58.98**| **46.55**| **51.48**| **52.77**| **54.68**|
> | 4:8      | SparseGPT     | 48.29    | 49.85    | 54.94    | 60.24    | 46.32  | 51.04  | 52.53  | 55.77   |
> | 4:8      | SDS-SparseGPT | **49.67**| **52.25**| **57.92**| **61.48**| **48.00**| **52.15**| **53.45**| **55.85**|
> | 4:8      | Wanda         | 46.28    | 46.41    | 55.04    | 58.21    | 44.63  | 49.15  | 51.17  | 54.18   |
> | 4:8      | SDS-Wanda     | **47.70**| **48.61**| **56.12**| **59.91**| **47.07**| **52.04**| **54.44**| **56.90**|
>
> The above table contains the experimental results with OPT/GPT as the base model and SparseGPT/Wanda as the base pruning method. Accuracy (%) is obtained by zero-shot evaluation and averaging over seven downstream tasks, including COPA, Lambada, OpenbookQA, PIQA, RTE, StoryCloze, and Winogrande.

---

> ### Author Response · Authors · 2023-11-22
> **Experimental results of using LLaMA as the basic to-prune PLM**
>
> | PLM          | Method        | Wiki.↓ | COPA↑ | BookQ.↑ | PIQA↑ | RTE↑  | Story.↑ | Wino.↑ | Avg(acc.)↑ |
> |--------------|---------------|--------|-------|---------|-------|-------|---------|--------|------------|
> | LLaMA-7b     | Dense         | 5.68   | 84    | 42.4    | 77.48 | 53.43 | 75.94   | 67.01  | 66.71      |
> | LLaMA-7b     | SparseGPT     | 11.23  | **80**| 35.4    | 70.62 | 59.57 | 69.89   | 59.98  | 62.58      |
> | LLaMA-7b     | SDS-SparseGPT | **9.97**| 79   | **35.6**| **71.65**| **59.57**| **71.67**| **61.80**| **63.22**|
> | LLaMA-7b     | Wanda         | 11.54  | 80    | 36.4    | 68.88 | **58.12**| 69.00   | 60.06  | 62.08      |
> | LLaMA-7b     | SDS-Wanda     | **9.73**| **82**| **36.8**| **70.57**| 57.60 | **70.27**| **62.04**| **63.21**|
> | LLaMA2-7b    | Dense         | 5.47   | 87    | 40.8    | 77.04 | 61.73 | 77.59   | 67.01  | 68.53      |
> | LLaMA2-7b    | SparseGPT     | 10.82  | 83    | 35.6    | 71.33 | 63.90 | 71.23   | 63.30  | 64.73      |
> | LLaMA2-7b    | SDS-SparseGPT | **10.02**| **84**| **37.2**| **71.44**| **64.09**| **71.99**| **63.88**| **65.43**|
> | LLaMA2-7b    | Wanda         | 12.14  | 79    | 34.6    | 70.08 | 59.57 | 69.70   | 60.22  | 62.20      |
> | LLaMA2-7b    | SDS-Wanda     | **10.39**| **81**| **36.8**| **70.62**| **64.25**| **71.8**| **63.06**| **64.59**|
>
> The pruning configuration is set to 2:4 sparsification.

---

### Official Review · Reviewer_k1aZ · 2023-11-01

**Soundness:** 3 good
**Presentation:** 3 good
**Contribution:** 2 fair
**Rating:** 5
**Confidence:** 4

**Summary:**

This paper builds on the SparseGPT work by Frantar and Alistarh, and proposes a Sparse-Dense-Sparse pruning framework to enhance the performance of pre-trained language models that have been pruned by just using the one-shot SparseGPT algorithm. The first sparse framework directly uses existing one-shot pruning algorithms, and the authors use SparseGPT during this phase. Then, a layer-wise knowledge distillation is applied using unlabeled training samples to recover the pruned connections in the model. The paper claims that the recovered dense model has enhanced pruning awareness for the subsequent pruning step. Finally, SparseGPT is applied again with weight adjusting to obtain the SDS sparse model which performs better than SparseGPT on smaller OPT models and on-par with SparseGPT on the larger models. The empirical performance is measured on raw-wikitext2 using perplexity and on some zero-shot downstream tasks like COPA, RTE, StoryCloze, Winogrande, etc.

**Strengths:**

- The paper tackles an important problem of sparsity in large language models that can help in reducing the memory footprint and reducing latency during inference for these large models.
- The paper is well written and is fairly easy to follow.
- The empirical results show gains over SparseGPT for the OPT-125m and OPT-350m models and on-par with SparseGPT for the OPT-1.3b and OPT-2.7b models (Table 2).

**Weaknesses:**

- Although the results on OPT models look okay on paper, I believe they are not enough to judge the practical relevance of the proposed method. First of all, how much additional flops are being incurred to prune the models in three stages? Secondly, the performance of OPT class of models on pre-training and various downstream tasks is itself not good. So, are the gains reported in the paper statistically significant, or they lie within the standard deviation of the performance of OPT models on these tasks.
- Sparsity and pruning research is more relevant for larger models to reduce their inference time and the GPU/TPU memory footprint. But the proposed method only matches SparseGPT's performance for the larger models. Is the further Dense-Sparse pruning even necessary?
- The paper should also report results on the speedup obtained compared to the dense and SparseGPT models with varying model size and sparsity category (50%, 2:4, 4:8).

**Questions:**

I have asked most of my questions in the weakness section, but here are a few more:

- Is there a typo in Algorithm 1 in section A.1? The main text of the paper mentions that it uses $W_{l}^{sparse-2nd}$ to collect $X_{l - 1}$ during forward propagation, but line 8 in the $\textbf{Second pruning: sparse weight adjustment}$ sub-algorithm mentions $W_{i}^{sparse}$ for forward propagation.
- I believe the authors use the same subset of unlabeled data during the second and third phases of SDS. Did the authors observe any difference in using a different sample during these two stages?
- Typo in Section 2.2, third last line: it should be unlabeled data and not labeled data, right? since only $X$ input is being used and intermediate outputs collected during forward propagation?

---

> ### Author Response · Authors · 2023-11-19
>
> Dear Reviewer k1aZ,
>
> Thank you for your incisive observations and questions regarding our work. We have addressed all your queries, offering comprehensive clarifications and supplementary data to reinforce our manuscript.
>
> **Response to weakness 1**: The process of running SDS is efficient and memory-friendly: (i) only the layer being optimized needs to be in GPU memory; (ii) optimization between layers is parallelizable, so the device's power can be fully utilized for speedups. The following table shows the runtime of SDS for single-device serial, multi-device (our implementation, on 8 32GB Nvidia V100 GPUs) parallel, and extreme parallel (theoretical) cases (see **Section A.5** for more analysis).
>
> |type|125M|1.3B|2.7B|7B|
> |--|--|--|--|--|
> |serial|~22min|~3.8hours|~7.4hours|~34.2hours|
> |parallel|~4min|~25min|~1hours|~4.5hours|
> |theoretical|<1min|~6min|~8min|~39min|
>
> To highlight the contrast in outcomes, we introduce additional experimental comparisons including GPT, LLaMA as the base model, and Wanda as the base pruning method, and calculated the average accuracy on all downstream tasks (**Table 6, Table 7, and Table 9 in Section A.5**). Experimental results demonstrate that SDS outperforms SparseGPT and Wanda by decreasing language comprehension perplexity by an average of 6.4 and achieving an average accuracy improvement of 1.9% across multiple downstream tasks. Thus, the SDS pruning method can be proven effective.
>
> *Additional comments are provided for your convenience to refer to the experimental results*.
>
> **Response to weakness 2**: In the revised manuscript, we included experiments involving the larger PLM LLaMA. Our findings demonstrate that SDS improves LLaMA's accuracy by 2.5% when compared to SparseGPT/Wanda. However, as the model size increases, redundancy also grows, allowing SparseGPT to achieve sufficiently good results independently. In such cases, SDS may not be necessary. However, for more recent models like LLaMa, which outperform OPT at similar sizes—highlighting LLaMA's compactness—SDS can still effectively enhance the performance of pruned PLMs.
>
> **Response to weakness 3**: Thanks for your interest in practical inference speedups for our pruned PLMs. Our approach is aimed at optimizing the performance of pruned PLMs under 50%/2:4/4:8 sparse configurations. The real-time speedup is contingent upon the support provided by sparse acceleration libraries or hardware capabilities. An instance of this is demonstrated by DeepSparse [1], showcasing a round 20% end-to-end speedup achieved through 50% unstructured sparsity on CPU. Nvidia [2] also demonstrates that a 2:4 sparse GEMM achieves an ideal twofold performance increase compared to an equivalent dense GEMM, leveraging Sparse Tensor Cores on GPUs with the Ampere architecture. Currently, 4:8 sparse acceleration is not supported. However, the experimental results in the manuscript will hold valuable implications for guiding future hardware designs.
>
> [1] DeepSparse: https://github.com/neuralmagic/deepsparse
>
> [2] Mishra A, Latorre J A, Pool J, et al. Accelerating sparse deep neural networks[J]. 2021.
>
> **Response to question 1**:  Thank you for pointing out the error in our formula. We have corrected it as follows: $X\leftarrow{W}_i^{\text{sparse-2nd}}X$ in the revised manuscript. We confirm that this correction does not impact our results or conclusions.
>
> **Response to question 2**: You are correct that we use the same samples throughout each SDS step. We appreciate your interest in using different samples within SDS, and the corresponding result is detailed in **row 11 of Table 4**. The efficacy of employing different samples within SDS closely mirrors that of the standard SDS, with only two tasks surpassing it. In the short term, this may be caused by the failure to learn sufficiently about each batch of samples as well as the varying quality of the samples. This finding confirms the advantage that SDS does not need to rely on additional samples.
>
> **Response to question 3**:  Yes, you are right. We intended to convey that our method utilizes only a small amount of unlabeled data. Acknowledging that the original phrasing (We did not use a large amount of labeled data) may have caused confusion, we have reworded the expression to be directly affirmative. We believe this change maintains the accuracy of our presentation while improving readability.
>
> **In summary**, we trust that our responses have thoroughly addressed your review points. Your valuable feedback has significantly helped refine our research. We welcome any further discussion or clarification if needed.
>
> Thank you for your invaluable contributions to our work.
>
> Best regards,
>
> Authors 5070.

---

> ### Author Response · Authors · 2023-11-22
> **Experimental results (Wikitext2 perplexity) of using Wanda as the basic pruning method**
>
> | Sparsity | Method           | OPT-125m | OPT-350m | OPT-1.3b | OPT-2.7b | GPT2-s | GPT2-m | GPT2-l | GPT2-xl |
> |----------|------------------|----------|----------|----------|----------|--------|--------|--------|---------|
> | 0        | Dense            | 27.66    | 22.01    | 14.62    | 12.16    | 29.95  | 21.71  | 19.33  | 17.41   |
> | 50%      | SparseGPT        | 36.85    | 31.58    | 17.46    | 13.48    | 47.46  | 28.06  | 23.55  | 19.91   |
> | 50%      | SDS-SparseGPT    | **34.23**| **29.36**| **17.40**| **13.42**| **45.76**| **27.91**| **23.32**| **19.62**|
> | 50%      | Wanda            | 39.79    | 41.88    | 18.36    | 14.38    | 46.71  | 29.29  | 24.89  | 20.83   |
> | 50%      | SDS-Wanda        | **35.05**| **33.07**| **17.23**| **13.74**| **40.32**| **27.39**| **23.15**| **19.78**|
> | 2:4      | SparseGPT        | 60.43    | 51.11    | 23.90    | 17.18    | 73.11  | 40.41  | 32.49  | 25.97   |
> | 2:4      | SDS-SparseGPT    | **51.30**| **46.23**| **23.02**| **17.36**| **64.31**| **38.24**| **31.33**| **25.05**|
> | 2:4      | Wanda            | 82.47    | 113.17   | 27.32    | 20.94    | 123.66 | 61.70  | 52.39  | 32.60   |
> | 2:4      | SDS-Wanda        | **59.17**| **73.56**| **23.94**| **18.02**| **63.57**| **41.11**| **31.11**| **25.35**|
> | 4:8      | SparseGPT        | 44.77    | 39.59    | 19.95    | 14.98    | 53.14  | 32.84  | 26.77  | 22.70   |
> | 4:8      | SDS-SparseGPT    | **41.66**| **34.18**| **19.54**| **14.81**| **50.90**| **32.41**| **26.29**| **22.27**|
> | 4:8      | Wanda            | 53.97    | 62.49    | 21.96    | 16.80    | 73.73  | 41.12  | 32.58  | 25.14   |
> | 4:8      | SDS-Wanda        | **43.58**| **47.31**| **19.82**| **15.45**| **51.05**| **33.55**| **26.16**| **22.11**|
>
> The above table contains the experimental results with OPT/GPT as the base model and SparseGPT/Wanda as the base pruning method.

---

> ### Author Response · Authors · 2023-11-22
> **Experimental results (average accuracy) of using Wanda as the basic pruning method**
>
> | Sparsity | Method        | OPT-125m | OPT-350m | OPT-1.3b | OPT-2.7b | GPT2-s | GPT2-m | GPT2-l | GPT2-xl |
> |----------|---------------|----------|----------|----------|----------|--------|--------|--------|---------|
> | 0        | Dense         | 50.82    | 54.12    | 60.83    | 62.81    | 50.07  | 53.58  | 56.51  | 58.40   |
> | 50%      | SparseGPT     | 48.85    | 52.33    | 55.89    | 61.14    | 47.27  | 52.82  | 53.47  | 57.22   |
> | 50%      | SDS-SparseGPT | **50.80**| **54.51**| **58.42**| **61.78**| **48.37**| **53.33**| **54.34**| **58.00**|
> | 50%      | Wanda         | 48.46    | 48.90    | 56.18    | 59.36    | 46.50  | 52.01  | 53.65  | 56.03   |
> | 50%      | SDS-Wanda     | **49.78**| **51.40**| **57.58**| **60.92**| **49.05**| **53.34**| **54.88**| **57.36**|
> | 2:4      | SparseGPT     | 47.56    | 48.34    | 53.57    | 58.48    | 46.47  | 50.17  | 50.85  | 53.57   |
> | 2:4      | SDS-SparseGPT | **49.59**| **50.50**| **56.67**| **59.96**| **47.62**| **50.65**| **52.45**| **56.25**|
> | 2:4      | Wanda         | 45.69    | 44.77    | 52.86    | 55.51    | 42.32  | 47.38  | 48.92  | 51.37   |
> | 2:4      | SDS-Wanda     | **47.09**| **46.69**| **54.44**| **58.98**| **46.55**| **51.48**| **52.77**| **54.68**|
> | 4:8      | SparseGPT     | 48.29    | 49.85    | 54.94    | 60.24    | 46.32  | 51.04  | 52.53  | 55.77   |
> | 4:8      | SDS-SparseGPT | **49.67**| **52.25**| **57.92**| **61.48**| **48.00**| **52.15**| **53.45**| **55.85**|
> | 4:8      | Wanda         | 46.28    | 46.41    | 55.04    | 58.21    | 44.63  | 49.15  | 51.17  | 54.18   |
> | 4:8      | SDS-Wanda     | **47.70**| **48.61**| **56.12**| **59.91**| **47.07**| **52.04**| **54.44**| **56.90**|
>
> The above table contains the experimental results with OPT/GPT as the base model and SparseGPT/Wanda as the base pruning method. Accuracy (%) is obtained by zero-shot evaluation and averaging over seven downstream tasks, including COPA, Lambada, OpenbookQA, PIQA, RTE, StoryCloze, and Winogrande.

---

> ### Author Response · Authors · 2023-11-22
> **Experimental results of using LLaMA as the basic to-prune PLM**
>
> | PLM          | Method        | Wiki.↓ | COPA↑ | BookQ.↑ | PIQA↑ | RTE↑  | Story.↑ | Wino.↑ | Avg(acc.)↑ |
> |--------------|---------------|--------|-------|---------|-------|-------|---------|--------|------------|
> | LLaMA-7b     | Dense         | 5.68   | 84    | 42.4    | 77.48 | 53.43 | 75.94   | 67.01  | 66.71      |
> | LLaMA-7b     | SparseGPT     | 11.23  | **80**| 35.4    | 70.62 | 59.57 | 69.89   | 59.98  | 62.58      |
> | LLaMA-7b     | SDS-SparseGPT | **9.97**| 79   | **35.6**| **71.65**| **59.57**| **71.67**| **61.80**| **63.22**|
> | LLaMA-7b     | Wanda         | 11.54  | 80    | 36.4    | 68.88 | **58.12**| 69.00   | 60.06  | 62.08      |
> | LLaMA-7b     | SDS-Wanda     | **9.73**| **82**| **36.8**| **70.57**| 57.60 | **70.27**| **62.04**| **63.21**|
> | LLaMA2-7b    | Dense         | 5.47   | 87    | 40.8    | 77.04 | 61.73 | 77.59   | 67.01  | 68.53      |
> | LLaMA2-7b    | SparseGPT     | 10.82  | 83    | 35.6    | 71.33 | 63.90 | 71.23   | 63.30  | 64.73      |
> | LLaMA2-7b    | SDS-SparseGPT | **10.02**| **84**| **37.2**| **71.44**| **64.09**| **71.99**| **63.88**| **65.43**|
> | LLaMA2-7b    | Wanda         | 12.14  | 79    | 34.6    | 70.08 | 59.57 | 69.70   | 60.22  | 62.20      |
> | LLaMA2-7b    | SDS-Wanda     | **10.39**| **81**| **36.8**| **70.62**| **64.25**| **71.8**| **63.06**| **64.59**|
>
> The pruning configuration is set to 2:4 sparsification.

---

> > ### Comment · Reviewer_k1aZ · 2023-11-22
> > **Response to Rebuttal**
> >
> > I thank the authors for providing a detailed response and additional experiments. The authors have addressed some of my concerns, therefore I am increasing my rating to 5. However, I cannot recommend for acceptance because the gains over simpler methods like SparseGPT are low and the paper does not report concrete latency numbers with the mentioned sparsity configurations.

---

### Official Review · Reviewer_hYrm · 2023-11-05

**Soundness:** 3 good
**Presentation:** 4 excellent
**Contribution:** 2 fair
**Rating:** 6
**Confidence:** 3

**Summary:**

The paper proposes a novel pruning framework called SDS (Sparse-Dense-Sparse) to enhance the performance of pruned Pre-trained Language Models (PLMs) while reducing computational and storage costs. SDS consists of three steps: initial pruning of less critical connections, reconstruction of a dense model with sparse regularization, and a second pruning round. The approach outperforms conventional one-shot pruning methods, such as SparseGPT, with limited calibration samples, achieving a decrease in language comprehension perplexity by 2.4 and an average accuracy improvement of over 2% across seven downstream tasks on OPTs.

**Strengths:**

- **Better performance than  SparseGPT**: Its performance seems better than SparseGPT.

- **Limited Calibration Samples**: SDS achieves superior results with a limited number of calibration samples, making it a practical and efficient approach for real-world applications where acquiring extensive labeled data might be challenging.

- **Detailed Process Explanation**: The paper provides a clear and detailed explanation of the three-step pruning process, enabling readers to understand the methodology thoroughly.

**Weaknesses:**

- It is unknown whether this could be valid for pruning large language models. For pruning small language models, there are already many solutions. I wonder the advantage of pruning.

**Questions:**

- why couldn't SDSDS or SDSD...SDS achieves better performance? In general, an  iterative SDS framework seems a good idea. Any ideas to make it work and find how many iteration to get the saturated performance.
- Can SDS also work for **large** language models? any insight?

---

> ### Author Response · Authors · 2023-11-19
>
> Dear Reviewer hYrm,
>
> Thank you for your valuable feedback on our submission. We appreciate the time you have dedicated to reviewing our work and offering constructive comments. Your insights are particularly helpful in understanding the implications of our pruning methods for both undersized and larger pre-trained language models and validating the impact of performing SDS optimization iteratively.
>
> In response to your concerns, we would like to provide some clarifications and insights based on our in-depth exploration and experiments.
>
> **Response to weakness 1**: We chose the Llama model to further validate the effectiveness of the SDS framework on larger and stronger PLMs. We found that when pruning the model using conventional pruning methods (SparseGPT/Wanda) incurs a large performance loss, SDS is effective in optimizing the pruned model. Below are the experimental results and analysis, you can also find them in the second part of Section A.5 (Page 19 in our updated manuscript).
>
> We take the 7b-sized models of LLaMA and LLaMA2 with 2:4 sparse as the pruning configuration and SparseGPT/Wanda as the base pruning method to verify the language comprehension perplexity and multi-task performance of the SDS-optimized pruned models, main results are shown below:
>
> |PLM|Method|Wiki.↓|COPA↑|BookQ.↑|PIQA↑|RTE↑|Story.↑|Wino.↑|Avg(acc.)↑|
> |--------------|-------------------|--------|-------|---------|-------|------|---------|--------|-------------|
> |Llama-7b|Dense|5.68|84|42.4|77.48|53.43|75.94|67.01|66.71|
> |Llama-7b|SparseGPT|11.23|80|35.4|70.62|59.57|69.89|59.98|62.58|
> |Llama-7b|SDS-SparseGPT|9.97|79|35.6|71.65|59.57|71.67|61.80|63.22|
> |Llama-7b|Wanda|11.54|80|36.4|68.88|58.12|69.00|60.06|62.08|
> |Llama-7b|SDS-Wanda|9.73|82|36.8|70.57|57.60|70.27|62.04|63.21|
> |Llama2-7b|Dense|5.47|87|40.8|77.04|61.73|77.59|67.01|68.53|
> |Llama2-7b|SparseGPT|10.82|83|35.6|71.33|63.90|71.23|63.30|64.73|
> |Llama2-7b|SDS-SparseGPT|10.02|84|37.2|71.44|64.09|71.99|63.88|65.43|
> |Llama2-7b|Wanda|12.14|79|34.6|70.08|59.57|69.70|60.22|62.20|
> |Llama2-7b|SDS-Wanda|10.39|81|36.8|70.62|64.25|71.80|63.06|64.59|
>
> The above results highlight that SDS achieves an average perplexity reduction of approximately 10.90% and an accuracy increase of around 2.46% over other one-shot pruning methods. This substantial improvement underscores the effectiveness of SDS in enhancing the performance of larger and stronger pruned PLMs.
>
> **Response to question 1**: The iterative implementation of the sparse-dense-sparse optimization failed to achieve better performance for the model, which we attribute to the overfitting phenomenon caused by overlearning the same batch or distribution of samples. In fact, optimizing with different samples in each iteration can produce higher-performance models. At the same time, we also found that SDS benefited more from multiple samples than SparseGPT.
>
> Specific experiments and more analysis can be found in Section A.4 (Page 17 in our updated manuscript).
>
> **Response to question 2**: SDS significantly enhances the performance of pruned undersized PLMs (OPTs) and larger PLMs (Llama). These models are characterized by low redundancy and they can be fully trained or trained on a sufficient number of samples. Conversely, models with notable redundancy, often comprising inadequately trained massive PLMs, can already be pruned by SparseGPT and Wanda with minimal or no loss. Consequently, the necessity to apply SDS and its optimization diminishes in such cases. SDS excels at efficiently optimizing pruned non-redundant PLMs, regardless of the model's volume (even though non-redundant PLMs often appear as undersized ones). As training techniques and high-quality data continue to be upgraded, newly born large-scale PLMs will increasingly exhibit low-redundancy characteristics, such as those of LLaMA. The SDS optimization will show more valid superiority ahead.
>
> **In summary**, we trust that our clarifications and additional data have effectively addressed your valuable feedback. We deeply appreciate the role your review has played in sharpening our research and contributing to its depth. We remain open to any additional insights or recommendations you may have.
>
> With gratitude for your thorough and constructive critique,
>
> Authors 5070.

---

> > ### Comment · Reviewer_XgFT · 2023-11-20
> > **Response**
> >
> > Thanks for providing the new results, and I think these results strongly demonstrate that the proposed approach is an effective additional step on top of any existing semi-structure pruning approaches. I would like to increase my score to 6.

---

### Meta-Review · Area_Chair_gzPQ · 2023-12-07

**Metareview:**

The paper proposes SDS, a Sparse-Dense-Sparse pruning framework that enhances the performance of pruned PLMs with a pruning-friendly weight distribution, requiring limited calibration samples and outperforming existing one-shot pruning methods in terms of language comprehension and accuracy across various tasks. This is a borderline paper and I apprecitate the authors' effort in improving the paper in the discussion period -- it is siginificantly improved. Hoever, after careful reading of the reviews, I tend to not accept it at the moment due to the following concerns. One reason for not accepting this paper at the moment is the insufficient level of novelty in the proposed method. In this field, there are already many simple yet effective compression techniques for large language models or lanugage models of smaller size. The approach presented in this work is incremental compared to the method in SparseGPT. Secondly, the improvement demonstrated by this method in the experiments is relatively insignificant, and it mainly is on models with smaller parameter sizes. Moreover, the paper lacks experimental results in many real-world applications for large language models, especially in text generation performance. These are the areas where the paper should be improved.

**Justification For Why Not Higher Score:**

n/a

**Justification For Why Not Lower Score:**

n/a

---

### Decision · Program_Chairs · 2024-01-16

Reject